# CAT Merging:
# A Training-Free Approach for Resolving Conflicts in Model Merging

Wenju Sun [1]  Qingyong Li [1]  Yangli-ao Geng [1]  Boyang Li [2]

## Abstract

Multi-task model merging offers a promising paradigm for integrating multiple expert models into a unified model without additional training. Existing state-of-the-art techniques, such as Task Arithmetic and its variants, merge models by accumulating task vectors—the parameter differences between pretrained and finetuned models. However, task vector accumulation is often hindered by knowledge conflicts, leading to performance degradation. To address this challenge, we propose **Conflict-Aware Task Merging (CAT Merging)**, a novel training-free framework that selectively trims conflict-prone components from the task vectors. CAT Merging introduces several parameter-specific strategies, including projection for linear weights and masking for scaling and shifting parameters in normalization layers. Extensive experiments on vision, language, and vision-language tasks demonstrate that CAT Merging effectively suppresses knowledge conflicts, achieving average accuracy improvements of up to 2.5% (ViT-B/32) and 2.0% (ViT-L/14) over state-of-the-art methods.

## 1. Introduction

Fine-tuning pretrained foundation models has become a standard paradigm for addressing downstream applications (Brown et al., 2020). However, as the number of applications grows, managing and deploying numerous finetuned models introduces significant costs and operational complexity. To address this issue, multi-task model merging has emerged as a promising solution by consolidating multiple

[1]Key Laboratory of Big Data & Artificial Intelligence in Transportation (Ministry of Education), School of Computer Science and Technology, Beijing Jiaotong University, Beijing, China [2]College of Computing and Data Science, Nanyang Technological University, Singapore. Correspondence to: Yangli-ao Geng <gengyla@bjtu.edu.cn>, Qingyong Li <liqy@bjtu.edu.cn>.

*Proceedings of the 42nd International Conference on Machine Learning*, Vancouver, Canada. PMLR 267, 2025. Copyright 2025 by the author(s).

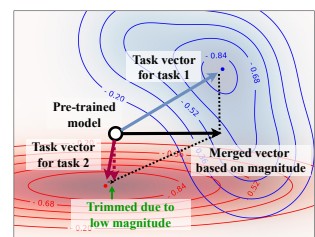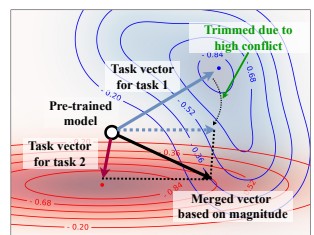

**Magnitude based Methods**          **CAT Merging (ours)**

*Figure 1.* Illustration of merging two conflicting task vectors. While masking low-magnitude components (the horizontal component of Task 2) mitigates some interference, high-magnitude dimensions (the vertical component of Task 1) can still induce significant conflicts. In contrast, the proposed CAT Merging proactively identifies and trims high-conflict components, stabilizing model merging.

expert models into a single unified model without additional training (Matena & Raffel, 2022).

A notable advancement in model merging is Task Arithmetic (Ilharco et al., 2023b), which introduces the concept of *task vectors*—defined as the difference vector between the pretrained and finetuned models in the parameter space. Task Arithmetic demonstrates that task-specific knowledge can be effectively integrated into the pretrained model through simple arithmetic operations, such as model merging by adding task vectors to the pretrained parameters. Despite its effectiveness, the technique is vulnerable to performance degradation caused by *knowledge conflict* (Sun et al., 2025; Ortiz-Jimenez et al., 2023), or the imbalance and contradiction among the accumulated task vectors.

To address knowledge conflict, prior research has explored enhancing Task Arithmetic by trimming unimportant components within task vectors. One typical importance metric is the parameter magnitude. For instance, Ties-Merging (Yadav et al., 2023) trims small-magnitude elements in task vectors. PCBMerging (Du et al., 2024) refines this importance measure by considering inter-task correlations. Similarly, Twin-Merging (Lu et al., 2024) applies singular value decomposition to task vectors; leveraging singular values as the importance metric, it retains components corresponding

to large singular values.

Despite the significant progress made by magnitude-based methods, they often overlook the risk of merging high-magnitude components that may overwrite task-specific knowledge during integration. As illustrated in Figure 1, methods like Ties-Merging prioritize dimensions with larger magnitudes (e.g., the vertical component of the Task 1 vector), which can unintentionally overwrite critical information from Task 2. This imbalance skews the merged model towards Task 1, degrading performance on Task 2 and undermining the overall multi-task capability. In contrast, the low-magnitude horizontal component of Task 2 carries valuable information while exerting minimal negative impact on Task 1. Trimming these components is therefore counterproductive. This example highlights the dual objectives of effective model merging: (1) suppressing conflicts where dominant knowledge from one task undermines the performance of others, and (2) preserving the unique and essential knowledge required by each task.

Motivated by the dual objectives outlined above, we propose Conflict-Aware Task Merging (CAT Merging), a feature-centric framework that addresses knowledge conflicts by trimming conflict-prone components from task vectors. Specifically, we focus on feature-level conflicts by analyzing task vector components layer by layer. By adhering to the dual objectives above, CAT Merging involves tailored operations for different types of parameters: feature projection for linear weights, and masking for normalization scalers and shifts. These strategies ensure that CAT Merging effectively mitigates knowledge conflicts in a training-free manner, relying solely on a lightweight forward pass with few unlabeled exemplars. We evaluate CAT Merging on diverse visual, language, and visual-language datasets, demonstrating its superiority over state-of-the-art methods while maintaining robustness with limited exemplars.

## 2. Related Work

### 2.1. Traditional Multi-Task Learning

Multi-task learning (MTL) seeks to enhance performance by transferring knowledge from related tasks (Zhang & Yang, 2022). In MTL, the major challenge is referred to as *negative transfer* (Liu et al., 2017; Zhang et al., 2023b), where the conflict knowledge among tasks results in degraded performance compared to training them independently. To address this, researchers have explored strategies such as sparsification (Ding et al., 2021; Sun et al., 2020; Liu et al., 2019a), modularization (Tang et al., 2020; Ma et al., 2018), and soft parameter sharing (Gao et al., 2020; Hazimeh et al., 2021). Others focus on the optimization process, such as dynamically weighting task-specific loss (Sener & Koltun, 2018; Liu et al., 2019a; 2022; Hu et al., 2023; Chen et al.,

2022), resolving gradient direction conflicts (Chen et al., 2020; Liu et al., 2021; Javaloy & Valera, 2022; Navon et al., 2022; Guo et al., 2022), or preventing over-prioritization of certain tasks (Chen et al., 2018; He et al., 2022; Yang et al., 2023).

### 2.2. Multi-Task Learning through Model Merging

Model merging has emerged as a promising approach for integrating knowledge across different models without retraining. Early techniques like Stochastic Weight Averaging (SWA) (Izmailov et al., 2018) averages model weights averaging during the final stages of training to enhance generalization. This concept was further advanced by approaches like SWAD (Cha et al., 2021) and Ensemble of Averages (EoA) (Arpit et al., 2022). Additionally, Ilharco et al. (2023a) empirically demonstrate that parameter averaging can effectively combine knowledge from models trained on diverse tasks. Based on this, Fisher-Merging (Matena & Raffel, 2022) applies the Fisher information matrix (Fisher, 1925) to weigh the averaging process. Similarly, RegMean (Jin et al., 2023) formulates model merging as the minimization of the distances between the merged model and individual finetuned models in the parameter space.

The pioneering work of Task Arithmetic (Ilharco et al., 2023b) introduces the concept of "task vectors"—parameter offsets of finetuned models from a pretrained model as the origin. By scaling and combining these vectors, Task Arithmetic effectively balances task-general and task-specific knowledge and achieves notable enhancement in model merging. Building on this idea, Ties-Merging (Yadav et al., 2023) and PCBMerging (Du et al., 2024) enhance the process by removing components in task vectors with negligible magnitudes. PEFT (Zhang et al., 2023a) and MoLE (Wu et al., 2024) integrate Task Arithmetic with LoRA modules (Hu et al., 2022) to enhance flexibility. Additionally, STA (Zhou et al., 2025) randomly shuffle network layers to eliminate knowledge conflict.

Some advanced approaches incorporate test-time training techniques to further improve merging performance. AdaMerging (Yang et al., 2024b) trains layer-specific merging coefficients. Additionally, representation surgery (Yang et al., 2024a) and MoE router (Tang et al., 2024) train an extra adapter to align representations. Since test-time adaptation requires additional training and samples, their application may be limited. Therefore, this paper focuses on training-free model merging.

## 3. Preliminary

**Notations.** Let $W_0 = \{W_0^l\}_{l=1}^{L}$ denote the parameters of a pretrained network with $L$ layers, where $W_0^l$ represents the parameters of layer $l$. The network is finetuned

independently on $K$ tasks using their respective datasets $D_1, \ldots, D_K$, yielding finetuned parameters $W_1, \ldots, W_K$. For a given task $k$, the task vector $T_k = \{T_k^l\}_{l=1}^L$ is defined as the difference between the finetuned parameters $W_k$ and the pretrained parameters $W_0$:

$$T_k = W_k - W_0 = \{W_k^l - W_0^l\}_{l=1}^L. \quad (1)$$

The dimensions of $W_0^l$ and $T_k^l$ vary depending on the parameter type in layer $l$. Specifically, if layer $l$ is a linear layer, then $W_0^l, T_k^l \in \mathbb{R}^{d_l \times d_{l+1}}$, where $d_l$ and $d_{l+1}$ are the input and output dimensions of layer $l$, respectively. For other parameters, such as element-wise scale or shift parameters (e.g., in layer normalization), $W_0^l, T_k^l \in \mathbb{R}^{d_l}$.

**Task Arithmetic.** Our objective is to merge $K$ finetuned models $\{W_k\}_{k=1}^K$ into a single multi-task model $W_{\text{mtl}}$ that performs well on all tasks. Task Arithmetic (Ilharco et al., 2023b) achieves this by simply adding task vectors to $W_0$:

$$W_{\text{mtl}} = W_0 + \alpha \sum_{k=1}^K T_k, \quad (2)$$

where $\alpha > 0$ is a manual scaling factor (we let $\alpha = 1$ for notational simplicity; our method allows for different $\alpha$ values). Despite its simplicity, this approach often suffers from knowledge conflicts when task vectors interfere, motivating our more refined trimming strategy.

## 4. Knowledge Conflict

Knowledge conflict (Sun et al., 2025) arises in model merging when task vectors are misaligned due to divergent magnitudes or directions. Direct aggregation, as employed in Task Arithmetic, often leads to performance degradation across tasks. We formally define knowledge conflict in Task Arithmetic as the increase in task-specific loss incurred by incorporating **other** task vectors. On a given task $k$, associated with the loss function $\mathcal{L}_k(.)$, the conflict introduced by another task vector $T_i$ ($i \neq k$) is quantified as:

$$\Delta \mathcal{L}_{k|i} = \mathcal{L}_k(W_k + T_i) - \mathcal{L}_k(W_k). \quad (3)$$

As mentioned, a task vector $T_i$ can be separated into network layers $T_i = \{T_i^1, \ldots, T_i^L\}$, where the superscript denotes the layer index. The effect of a perturbation $T_i$ propagates through the layers. Without loss of generality, we suppose all layers are sequentially stacked, i.e., the input to layer $l$ is the output of layer $l - 1$, where the $l$-th layer feature on task $k$, denoted as $f_k^l(.)$, can be written as:

$$f_k^l(W_k + T_i) = f_k^l(f_k^{l-1}(W_k + T_i); W_k^l + T_i^l). \quad (4)$$

To isolate the local effect of $T_i^l$, we define:

$$\hat{f}_k^l(W_k + T_i) = f_k^l(f_k^{l-1}(W_k); W_k^l + T_i^l), \quad (5)$$

where the input features are computed using only the unperturbed parameters $\{W_k^1, \ldots, W_k^{l-1}\}$. This allows the total feature shift at layer $l$, $\Delta f_{k|i}^l = f_k^l(W_k + T_i) - f_k^l(W_k)$, to be controlled under a mild Lipschitz assumption:

**Assumption 4.1** (Lipschitz Continuity of Network Layers). On the line segment between $W_k$ and $W_k + T_i$, the function of layer $l$, denoted as $f_k^l$, is $\gamma_l$-Lipschitz continuous with respect to its input (i.e., the output of layer $l - 1$).

**Lemma 4.2.** *By Assumption 4.1, we have:*

$$\|\Delta f_{k|i}^l\| \leq \gamma_l \|\Delta f_{k|i}^{l-1}\| + \|\Delta \hat{f}_{k|i}^l\|, \quad (6)$$

*where $\Delta \hat{f}_{k|i}^l = \hat{f}_k^l(W_k + T_i) - f_k^l(W_k)$.*

Intuitively, Lemma (4.2) decomposes the feature shift at layer $l$ into (i) the propagated feature shift $\Delta f_{k|i}^{l-1}$ from the previous layer and (ii) the **layer-wise** parameter perturbation $\Delta \hat{f}_{k|i}^l$ caused by $T_i^l$. We refer the reader to Section A.1 for the detailed proof.

We further consider how perturbations in the features extracted from the network affect the task-specific loss and impose a continuity condition on the loss function:

**Assumption 4.3** (Lipschitz Continuity of Loss Function). On the line segment between $W_k$ and $W_k + T_i$, the loss function $\mathcal{L}$ is $\beta$-Lipschitz continuous with respect to the final output of the network.

Combining the above, we have

**Theorem 4.4** (An Upper Bound on Knowledge Conflict). *Suppose that Assumptions 4.1 and 4.3 hold for any task $k$ and task vector $T_i$, the knowledge conflict follows*

$$|\Delta \mathcal{L}_{k|i}| \leq \beta \sum_{l=1}^L \Big( \prod_{m=l+1}^L \gamma_m \Big) \|\Delta \hat{f}_{k|i}^l\|. \quad (7)$$

The proof is provided in Section A.2. Theorem 4.4 suggests that to reduce the upper bound on knowledge conflict, we can directly minimize the layer-wise perturbations $\|\Delta \hat{f}_{k|i}^l\|$ for each layer. In the next section, we develop a practical merging algorithm to do so.

## 5. Methodology

As introduced in Section 4, our approach adopts a layer-wise strategy to analyze and mitigate knowledge conflict locally at each layer. Several existing methods attempt to reduce conflict by preserving components deemed important to each task. For example, Fisher Merging (Matena & Raffel, 2022) utilizes the Fisher Information Matrix to retain task-important weights, while Ties Merging (Yadav et al., 2023) and PCB Merging (Du et al., 2024) prune low-magnitude parameters.

Although suppressing unimportant parameters can reduce some conflicts, these approaches overlook a critical issue: high-magnitude parameters—often considered important—can themselves be major sources of knowledge conflict. These components may overpower others during merging, distorting or suppressing the feature representations vital to different tasks. As a result, even when low-magnitude parameters are pruned, significant knowledge conflict still exists, degrading multi-task performance.

In contrast to existing trimming techniques, we propose a method that explicitly balances two objectives: (1) minimizing interference between tasks and (2) preserving the knowledge encoded in task vectors. For each task $k$, we define the task-specific transformation $\Phi_k$ that trims other task vectors in the same layer, i.e., $T_i$ ($i \neq k$). The objective for $\Phi_k(.)$ is to minimize the layer-wise knowledge conflict $\Delta f_{k,i}^l$ while maintaining the representational integrity of $T_i$. Formally, considering all task vectors yields the following objective for $\Phi_k(.)$:

$$
\min_{\Phi_k} \sum_{i \neq k} \underbrace{\left\| \hat{f}_k\left(W_k + \Phi_k(T_i)\right) - f_k(W_k) \right\|^2}_{\text{Inter-task Knowledge Conflict } \Delta \hat{f}_{k|i}^l}
$$
$$
+ \lambda \underbrace{\left\| \hat{f}_i(W_0 + \Phi_k(T_i)) - f_i(W_i) \right\|^2}_{\text{Intra-task Knowledge Deviation}}, \quad (8)
$$

where $\lambda$ controls the trade-off between reducing conflict and preserving knowledge.

Based on the computational rules of different parameter types, we design tailored trimming operations. Specifically, for modern transformer-based networks, the parameters can be broadly classified into three categories:

- Weight parameters in linear layers, which involve matrix multiplication operations;

- Scale parameters in normalization layers, which scale normalized features through element-wise Hadamard products;

- Shift parameters, which modify features through element-wise addition.

Most parameters in modern architectures fall into these categories. Notably, convolutional weights are treated analogously to linear weights, since convolution can be expressed as a linear operation (Sun et al., 2023). Furthermore, complex modules such as attention blocks can be decomposed into multiple linear transformations—e.g., for query, key, and value projections—allowing the trimming strategy to be applied independently to each component. In the following section, we provide detailed layer-wise implementations of $\Phi_k(.)$ tailored to each parameter type.

## 5.1. Linear Weight Trimming

Linear layers transform features via matrix multiplication. Given $X_k \in \mathbb{R}^{n \times d_l}$ extracted by former layers $\{W_k^1, \ldots, W_k^{l-1}\}$, the weight matrix $W \in \mathbb{R}^{d_l \times d_{l+1}}$ produce the output as follows:

$$
f(X_k; W) = X_k W. \quad (9)
$$

To mitigate knowledge conflicts at this layer, we aim to remove interfering components from other task vectors $T_i, i \neq k$. We introduce a **removal** basis $B_k \in \mathbb{R}^{d_{l+1} \times c}$, representing a $c$-dimensional core sub-parameter space for task $k$. For any $i \neq k$, we define a projection operator $\Phi_k(T_i) = T_i - T_i B_k B_k^\top$ which discards components of $T_i$ aligned with $B_k$, preserving task-specific knowledge while minimizing interference with task $k$.

Using the layer-wise loss from Eq. (8), we formulate the optimization objective for $B_k$ as:

$$
\sum_{i \neq k} \left\| X_k\left(W_0 + T_k + T_i - T_i B_k B_k^\top\right) - X_k(W_0 + T_k) \right\|_F^2
$$
$$
+ \lambda \sum_{i \neq k} \left\| X_i(W_0 + T_i - T_i B_k B_k^\top) - X_i(W_0 + T_i) \right\|_F^2
$$
$$
= \sum_{i \neq k} \left( \left\| X_k T_i - X_k T_i B_k B_k^\top \right\|_F^2 + \lambda \left\| X_i T_i B_k B_k^\top \right\|_F^2 \right). \quad (10)
$$

Applying the identity $\|X_k T_i\|_F^2 = \left\| X_k T_i B_k B_k^\top \right\|_F^2 + \left\| X_k T_i - X_k T_i B_k B_k^\top \right\|_F^2$ (see Appendix A.3 for the proof), we recast the objective as a maximization problem:

$$
\max_{B_k} \sum_{i \neq k} \left( \left\| X_k T_i B_k B_k^\top \right\|_F^2 - \lambda \left\| X_i T_i B_k B_k^\top \right\|_F^2 \right). \quad (11)
$$

The optimal removal basis $B_k$ is constructed from the top-$c$ eigenvectors of the matrix: $\sum_{i \neq k} T_i^\top (X_k^\top X_k - \lambda X_i^\top X_i) T_i$, where the full derivation is provided in Appendix A.4. Here, $c$ is a manually defined dimensionality hyperparameter. A higher value of $c$ corresponds to the removal of more information. In low-sample regimes, we recommend setting $c = 2$ or 3 to avoid introducing noise.

## 5.2. Scaling Parameter Trimming

Normalization layers are crucial for stabilizing the training of modern neural networks. Although normalization itself is a parameter-free operation, widely used layers such as batch normalization and layer normalization typically include an affine transformation, which consists of scale and shift parameters. In this section, we focus on merging the

scale parameters across tasks. At a normalization layer $l$, the scaler $W \in \mathbb{R}^{d_l}$ performs element-wise multiplication on the normalized feature $x_k \in \mathbb{R}^{d_l}$:

$$f(x_k; W) = x_k \circ W, \qquad (12)$$

where $\circ$ denotes element-wise (Hadamard) multiplication.

To reduce interference at this layer, we introduce a binary **removal** mask $m_k \in \{0,1\}^{d_l}$ applied to task vectors $T_i$ ($i \neq k$), such that the refined task vector $\Phi_k(T_i) = T_i - T_i \circ m_k$ has minimal influence on task $k$. Adapting the layer-wise loss from Eq. (8), we define the objective:

$$\sum_{i \neq k} \sum_{x_k} \left\| x_k \circ \left( W_0 + T_k + T_i - T_i \circ m_k \right) - x_k \circ (W_0 + T_k) \right\|^2$$

$$+ \lambda \sum_{x_i} \| x_i \circ (W_0 + T_i - T_i \circ m_k) - x_i \circ (W_0 + T_i) \|^2$$

$$= \sum_{i \neq k} \left( \sum_{x_k} \| x_k \circ T_i - x_k \circ T_i \circ m_k \|^2 + \lambda \sum_{x_i} \| x_i \circ T_i \circ m_k \|^2 \right).$$

$$(13)$$

Using the identity $\| x_k \circ T_i - x_k \circ T_i \circ m_k \|^2 + \| x_k \circ T_i \circ m_k \|^2 = \| x_k \circ T_i \|^2$, we reformulate the problem as a maximization objective:

$$\max_{m_k} \sum_{i \neq k} \left( \sum_{x_k} \| x_k \circ T_i \circ m_k \|^2 - \lambda \sum_{x_i} \| x_i \circ T_i \circ m_k \|^2 \right). \qquad (14)$$

This motivates selecting the top-$c$ dimensions in $m_k$ corresponding to the largest components of the vector: $\sum_{i \neq k} \left( \sum_{x_k} (x_k \circ T_i)^2 - \lambda \sum_{x_i} (x_i \circ T_i)^2 \right)$, with remaining entries set to 0. The selection of $c$ is match the number of basis $B_k$ used in the previous linear weight trimming to reduce noise from limited exemplars. Full mathematical derivations are provided in Appendix A.5.

### 5.3. Shifting Parameter Trimming

Shift parameters $W \in \mathbb{R}^{d_l}$ adjust features via element-wise addition, the transformation applied to input $x_k \in \mathbb{R}^{d_l}$ is defined as:

$$f(x_k; W) = x_k + W. \qquad (15)$$

We also adopt a binary **removal** mask $m_k \in \{0,1\}^{d_l}$, yielding a refined vector $\Phi_k(T_i) = T_i - T_i \circ m_k$. The corresponding objective function becomes:

---

**Algorithm 1** The model merging process

**Input:** Pretrained model $W_0$; Task vectors $\{T_1, \ldots, T_K\}$; Unlabeled exemplar-set $\{M_1, \ldots, M_K\}$

**Output:** Merged model $W_{\text{mtl}}$

1 **// Collecting the input features**
  **for** $k = 1$ *to* $K$ **do**
2     Initialize task inputs: $X_k^1 = M_k$
      **for** $l = 1$ *to* $L$ **do**
3       $\lfloor\ X_k^{l+1} = f(X_k^l; W_0^l + T_k^l)$

4 **// Compute basis or mask**
  **for** $k = 1$ *to* $K$; $l = 1$ *to* $L$ **do**
5     $\Phi_k^l = \arg\min_{\Phi_k^l} \sum_{i \neq k}$
      $\| f_k^l \left( W_0^l + T_k^l + \Phi_k^l(T_i^l) \right) - f_k^l(W_0^l + T_k^l) \|^2$
      $+ \lambda \| f_i^l(W_0^l + \Phi_k^l(T_i^l)) - f_i^l(W_0^l + T_i^l) \|^2$, // Eq. (8)

6 **// Edit all task vectors**
  **for** $k = 1$ *to* $K$; $l = 1$ *to* $L$; $i = 1$ *to* $K$, $i \neq k$ **do**
7     $\lfloor\ T_i^l = \Phi_k^l(T_i^l)$

8 **// Merging**
  $W_{\text{mtl}} = W_0 + \alpha \sum_k T_k$
  **return** $W_{\text{mtl}}$

---

$$\sum_{i \neq k} \sum_{x_k} \| x_k + (W_0 + T_k + T_i - T_i \circ m_k) - x_k - (W_0 + T_k) \|^2$$

$$+ \lambda \sum_{x_i} \| x_i + (W_0 + T_i - T_i \circ m_k) - x_i - (W_0 + T_i) \|^2$$

$$= \sum_{i \neq k} \left( \sum_{x_k} \| T_i - T_i \circ m_k \|^2 + \lambda \sum_{x_i} \| T_i \circ m_k \|^2 \right).$$

$$(16)$$

Similarly, the mask $m_k$ can be determined by a maximization problem:

$$\max_{m_k} \sum_{i \neq k} \left( \sum_{x_k} \| T_i \circ m_k \|^2 - \lambda \sum_{x_i} \| T_i \circ m_k \|^2 \right). \qquad (17)$$

Assuming data is approximately balanced across tasks, the optimal mask $m_k$ is obtained by selecting the top-$c$ components of $\sum_{i \neq k} (T_i)^2$ and setting the corresponding entries of $m_k$ to 1, with the rest set to 0. Full derivations are provided in Appendix A.6.

We have now finished discussing the trimming strategy for all major parameter types. During the merging process, we start with a layer-by-layer forward pass using only a small number of unlabeled samples—typically two or three per task—to collect necessary inputs for parameter trimming. The collected inputs are then processed to compute the corresponding basis or masks for editing task vectors based on the parameter type. The computed basis or masks are

*Table 1.* Multi-task performance when merging ViT-B/32 models on eight vision tasks. The best and second-best performances are written in **bold** and underlined text. The "#best" column represents the number of datasets where the method performs the best.

| Method | SUN397 | Cars | RESISC45 | EuroSAT | SVHN | GTSRB | MNIST | DTD | Avg Acc | #best |
|---|---|---|---|---|---|---|---|---|---|---|
| Pretrained | 62.3 | 59.7 | 60.7 | 45.5 | 31.4 | 32.6 | 48.5 | 43.8 | 48.0 | - |
| Individual | 75.3 | 77.7 | 96.1 | 99.7 | 97.5 | 98.7 | 99.7 | 79.4 | 90.5 | - |
| Traditional MTL | 73.9 | 74.4 | 93.9 | 98.2 | 95.8 | 98.9 | 99.5 | 77.9 | 88.9 | - |
| Weight Averaging | 65.3 | 63.4 | 71.4 | 71.7 | 64.2 | 52.8 | 87.5 | 50.1 | 65.8 | 0 |
| Fisher Merging | **68.6** | **69.2** | 70.7 | 66.4 | 72.9 | 51.1 | 87.9 | 59.9 | 68.3 | 2 |
| RegMean | 65.3 | 63.5 | 75.6 | 78.6 | 78.1 | 67.4 | 93.7 | 52.0 | 71.8 | 0 |
| Task Arithmetic | 55.2 | 54.9 | 66.7 | 78.9 | 80.2 | 69.7 | 97.3 | 50.4 | 69.1 | 0 |
| Ties-Merging | 59.8 | 58.6 | 70.7 | 79.7 | 86.2 | 72.1 | 98.3 | 54.2 | 72.4 | 0 |
| TATR | 62.7 | 59.3 | 72.3 | 82.3 | 80.5 | 72.6 | 97.0 | 55.4 | 72.8 | 0 |
| Ties-Merging & TATR | 66.3 | 65.9 | 75.9 | 79.4 | 79.9 | 68.1 | 96.2 | 54.8 | 73.3 | 0 |
| Consensus Merging | 65.7 | 63.6 | 76.5 | 77.2 | 81.7 | 70.3 | 97.0 | 57.1 | 73.6 | 0 |
| PCB Merging | 63.8 | 62.0 | 77.1 | 80.6 | **87.5** | **78.5** | **98.7** | 58.4 | 75.8 | 3 |
| CAT Merging (ours) | 68.1 | 65.4 | **80.5** | **89.5** | 85.5 | **78.5** | 98.6 | **60.7** | **78.3** | 4 |

*Table 2.* Multi-task performance when merging ViT-L/14 models on eight vision tasks.

| Method | SUN397 | Cars | RESISC45 | EuroSAT | SVHN | GTSRB | MNIST | DTD | Avg Acc | #best |
|---|---|---|---|---|---|---|---|---|---|---|
| Pretrained | 66.8 | 77.7 | 71.0 | 59.9 | 58.4 | 50.5 | 76.3 | 55.3 | 64.5 | - |
| Individual | 82.3 | 92.4 | 97.4 | 100.0 | 98.1 | 99.2 | 99.7 | 84.1 | 94.2 | - |
| Traditional MTL | 80.8 | 90.6 | 96.3 | 96.3 | 97.6 | 99.1 | 99.6 | 84.4 | 93.5 | - |
| Weight Averaging | 72.1 | 81.6 | 82.6 | 91.9 | 78.2 | 70.7 | 97.1 | 62.8 | 79.6 | 0 |
| Fisher Merging | 69.2 | **88.6** | 87.5 | 93.5 | 80.6 | 74.8 | 93.3 | 70.0 | 82.2 | 1 |
| RegMean | 73.3 | 81.8 | 86.1 | **97.0** | 88.0 | 84.2 | 98.5 | 60.8 | 83.7 | 1 |
| Task Arithmetic | 73.9 | 82.1 | 86.6 | 94.1 | 87.9 | 86.7 | 98.9 | 65.6 | 84.5 | 0 |
| Ties-Merging | 76.5 | 85.0 | 89.3 | 95.7 | 90.3 | 83.3 | 99.0 | 68.8 | 86.0 | 0 |
| TATR | 74.6 | 83.7 | 87.6 | 93.7 | 88.6 | 88.1 | 99.0 | 66.8 | 85.3 | 0 |
| Ties-Merging & TATR | 76.3 | 85.3 | 88.8 | 94.4 | 90.8 | 88.7 | 99.2 | 68.8 | 86.5 | 0 |
| Consensus Merging | 75.0 | 84.3 | 89.4 | 95.6 | 88.3 | 82.4 | 98.9 | 68.0 | 85.2 | 0 |
| PCB Merging | 76.2 | 86.0 | 89.6 | 95.9 | 89.9 | 92.3 | 99.2 | 71.4 | 87.6 | 0 |
| CAT Merging (ours) | **78.7** | 88.5 | **91.1** | 96.3 | **91.3** | 95.7 | **99.4** | 75.7 | **89.6** | 6 |

applied to all task vectors, enabling the final merging in Eq. (2) with minimal knowledge conflict. The detailed implementation of this process is described in Algorithm 1.

To simplify analysis, we have assumed the scaling factor $\alpha$ (cf. Eq. (2))to be 1 in the above analysis, but setting $\alpha$ to another positive value does not affect the resulting conclusion. For example, in Eqs. (10) & (11), by explicitly including $\alpha$, the optimal vector $B_k$ corresponds to the eigenvector of the matrix: $\sum_{i \neq k} (\alpha T_i^\top)(X_k^\top X_k - \lambda X_i^\top X_i)(\alpha T_i) = \alpha^2 \sum_{i \neq k} T_i^\top (X_k^\top X_k - \lambda X_i^\top X_i)T_i$. Since scaling a matrix by a positive constant $\alpha^2$ does not alter its eigenvectors, our theoretical conclusions remain valid. In practice, $\alpha$ is a critical hyperparameter that must be tuned carefully.

# 6. Experiments

## 6.1. Settings

**Datasets.** We select diverse datasets to evaluate our work, including eight vision datasets: SUN397 (Xiao et al., 2016), Cars (Krause et al., 2013), RESISC45 (Cheng et al., 2017), EuroSAT (Helber et al., 2019), SVHN (Netzer et al., 2011), GTSRB (Stallkamp et al., 2011), MNIST (LeCun & Cortes, 2010), DTD (Cimpoi et al., 2014), six visual-language datasets: COCO Caption (Chen et al., 2015), Flickr30k Caption (Plummer et al., 2015), Textcaps (Sidorov et al., 2020), OKVQA (Marino et al., 2019), TextVQA (Singh et al., 2019), and ScienceQA (Lu et al., 2022), and eight NLP tasks in the GLUE benchmark (Wang et al., 2019).

**Baselines.** We compare our proposed method with various training-free model merging methods, including the straightforward Weight Average, Fisher Merging (Matena & Raffel,

*Table 3.* Multi-task performance when merging RoBERTa models on eight NLP tasks.

| Method | CoLA | MNLI | MRPC | QNLI | QQP | RTE | SST2 | STS-B | Average | #best |
|---|---|---|---|---|---|---|---|---|---|---|
| Task Arithmetic | 6.68 | 66.23 | **78.46** | 78.62 | 72.69 | 53.43 | 83.49 | 27.10 | 58.34 | 1 |
| Ties-Merging | 9.46 | 59.34 | 74.71 | 65.93 | 41.29 | 47.29 | 72.13 | 9.21 | 47.42 | 0 |
| TATR | 10.20 | 65.44 | 72.56 | 75.73 | 74.58 | 55.18 | 78.87 | 37.46 | 58.39 | 0 |
| PCB Merging | 11.40 | 50.85 | 77.63 | 78.22 | 55.78 | 60.29 | 75.57 | **67.01** | 59.59 | 1 |
| CAT Merging (ours) | **33.20** | **72.33** | 68.22 | **82.92** | 76.05 | 62.82 | **89.33** | 15.57 | **62.56** | **6** |

*Table 4.* Multi-task performance when merging BLIP models on six vision-language tasks.

| Method | COCO Caption | Flickr30k Caption | Textcaps | OKVQA | TextVQA | ScienceQA | #best |
|---|---|---|---|---|---|---|---|
| | CIDEr | CIDEr | CIDEr | Accuracy | Accuracy | Accuracy | |
| Pretrained | 0.07 | 0.03 | 0.05 | 42.80 | 21.08 | 40.50 | - |
| Task Arithmetic | 0.86 | 0.50 | **0.39** | 17.71 | 0.49 | 40.10 | 1 |
| Ties-Merging | 0.53 | 0.27 | 0.22 | 27.95 | 0.57 | 40.35 | 0 |
| TATR | 0.46 | 0.31 | 0.21 | 28.30 | 14.74 | 42.98 | 0 |
| PCB Merging | 0.71 | 0.52 | 0.30 | 36.04 | 1.88 | 43.01 | 0 |
| CAT Merging (ours) | **0.91** | **0.53** | 0.36 | **44.07** | **19.69** | **46.36** | 5 |

2022), RegMean (Jin et al., 2023), Task Arithmetic (Ilharco et al., 2023b), Ties-Merging (Yadav et al., 2023), Task Arithmetic in Trust Region (TATR) (Sun et al., 2025), Consensus Merging (Wang et al., 2024), and PCB Merging (Du et al., 2024). We also provide the performance of some baseline methods, such as the pretrained model, the individual task model, and the traditional multi-Task learning model.

**Implementation details.** Our implementation strictly follows Task Arithmetic (Ilharco et al., 2023b). For the vision tasks, we employ the vision encoder from CLIP (Radford et al., 2021) as the pretrained backbone, experimenting with both ViT-B/32 and ViT-L/14 architectures. Task vectors are directly sourced from the repository provided by Ilharco et al. (2023b). After merging the models, we evaluate and report the accuracy of each task, along with the overall average accuracy. For the visual-language tasks, we derive task vectors by finetuning the visual question-answering (VQA) version of BLIP (Li et al., 2022), training each task for 6,000 steps. The architecture of BLIP includes an image encoder, a text encoder, and a text decoder, with all components finetuned during training. We use CIDEr as the evaluation metric for caption tasks and accuracy for VQA tasks. For the NLP tasks, we use RoBERTa (Liu et al., 2019b) as the backbone. Task vectors are obtained from Lu et al. (2024). Following their setting, we reserve 10% of the training set for validation and employ the original validation data as the test set.

## 6.2. Comparison Results

**Evaluation on vision tasks.** Tables 1 and 2 present the merging results on eight image classification benchmarks. CAT Merging achieves state-of-the-art performance, with

average accuracies of 78.3% on ViT-B/32 and 89.6% on ViT-L/14. Notably, substantial improvements are observed on challenging datasets such as EuroSAT (89.5%, ViT-B/32) and GTSRB (95.7%, ViT-L/14). Additional results for ViT-B/16 and comparisons with test-time training methods are provided in the supplementary materials.

**Evaluation on NLP tasks.** We evaluate CAT Merging on eight diverse NLP tasks using RoBERTa, covering both classification and regression (e.g., STS-B). For classification, we report accuracy, while for regression we report the mean of Pearson and Spearman correlations. As summarized in Table 3, CAT Merging consistently outperforms baseline methods, achieving the best results on six out of eight tasks, demonstrating strong generalization and robustness in model merging for language tasks.

**Evaluation on visual-language tasks.** Table 4 reports results using the BLIP model on six vision-language tasks that potentially conflict with each other (Tiong et al., 2024). CAT Merging achieves top performance on five of six benchmarks, highlighting its effectiveness in multimodal tasks.

## 6.3. Ablation Study

This section performs an ablation study to evaluate the effectiveness of the parameter-specific trimming techniques in CAT Merging, including the trimming of linear weights, normalization scalers, and shift parameters. Table 5 presents the results with ViT-B/32 and ViT-L/14 architectures.

Excluding the trimming of linear weights leads to a significant performance drop, with average accuracies of 71.5% (ViT-B/32) and 85.6% (ViT-L/14). Similarly, removing the trimming of normalization scalers and shifts also causes

*Table 5.* An ablation study of CAT Merging, where we selectively trim three different types of network parameters.

| Trimmed parameters | | | Architecture | |
|---|---|---|---|---|
| Linear weight | Normalization scaler | Shift | ViT-B/32 | ViT-L/14 |
| $\times$ | $\checkmark$ | $\checkmark$ | 71.5 | 85.6 |
| $\checkmark$ | $\times$ | $\checkmark$ | 76.8 | 89.2 |
| $\checkmark$ | $\checkmark$ | $\times$ | 77.8 | 88.7 |
| $\checkmark$ | $\checkmark$ | $\checkmark$ | **78.3** | **89.6** |

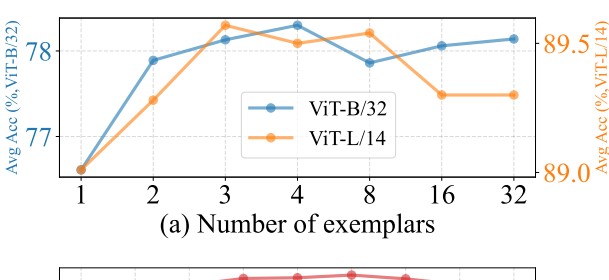

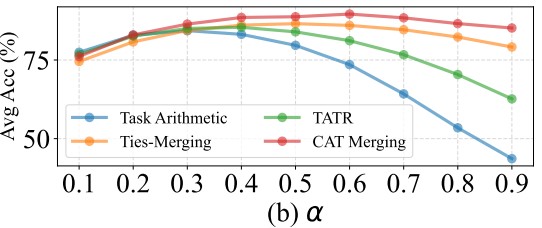

*Figure 2.* (a) Average accuracy (%) of CAT Merging on eight vision tasks with different numbers of exemplars per task. (b) Average accuracy (%) on eight vision tasks with ViT/L-14 models versus different $\alpha$ (scaling factor in Task Arithmetic, cf. Eq. (2)).

performance degradation. With all three types trimmed, we achieve the highest average accuracy. These findings highlight the importance of mitigating knowledge conflict in all types of parameters.

### 6.4. Sensitivity Analysis

**Sensitivity analysis of exemplar number.** As shown in Figure 2 (a), CAT Merging is robust to the number of exemplars. It achieves strong results even with just one exemplar per task (76.61% for ViT-B/32 and 89.01% for ViT-L/14). This is likely due to the patch-based design of ViT, which splits one image into several patches, ensuring the diversity of each layer input even with limited samples.

**Sensitivity analysis of $\alpha$.** In Figure 2 (b), with different values of $\alpha$, CAT Merging achieves more stable performance than Task Arithmetic. The stability should be attributed to suppressing knowledge conflict among task vectors, analyzed further in the next section. A detailed analysis of other

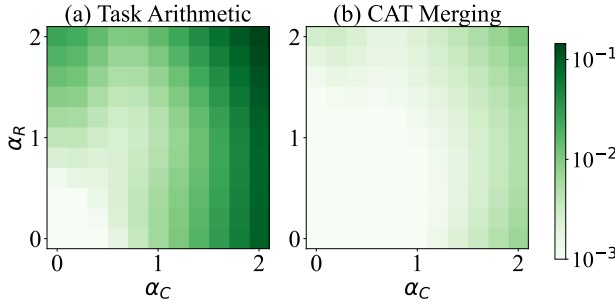

*Figure 3.* Visualization of knowledge conflict on Cars and RE-SISC45 (i.e., $\Delta L_{\text{Cars,RESISC45}} + \Delta L_{\text{RESISC45,Cars}}$) when merging ViT-L/14 models with different merging weights $\alpha_C$ and $\alpha_R$.

hyperparameters, including weight $\lambda$ and $c$, is provided in Appendix C.4.

### 6.5. Analysis of Knowledge Conflict

This section investigates the effectiveness of CAT Merging in mitigating knowledge conflict. Specifically, we consider the task vectors corresponding to Cars and RE-SISC45 and merge them using distinct scaling factors, i.e., $W_{\text{mtl}} = W_0 + \alpha_C T_{\text{Cars}} + \alpha_R T_{\text{RESISC45}}$. Figure 3 visualizes the knowledge conflict during merging under different merging coefficients. For Task Arithmetic, regions with minimal knowledge conflict are primarily concentrated where both $\alpha_C$ and $\alpha_R$ are close to 0. In contrast, CAT Merging effectively reduces knowledge conflict, enabling the use of much larger $\alpha$ values.

## 7. Conclusion

In this work, we propose CAT Merging, a novel training-free technique for multi-task model merging. By introducing parameter-specific strategies, such as projection for linear weights and masking for normalization and shift parameters, CAT Merging effectively resolves inter-task knowledge conflicts while preserving task-specific information. Experimental results across diverse vision, language, and vision-language tasks demonstrate the effectiveness of CAT Merging. We achieve significant accuracy improvements of up to 2.5% on ViT-B/32 and 2.0% on ViT-L/14 compared to existing methods. CAT Merging also exhibits strong robustness in the case with only a few exemplars, providing a practical solution for real-world multi-task scenarios where retraining is infeasible.

## Impact Statement

This paper presents work whose goal is to advance the field of Machine Learning. There are many potential societal

consequences of our work, none which we feel must be specifically highlighted here.

## Acknowledgments

This work was supported, in part, by the Fundamental Research Funds for the Central Universities under Grants 2023JBZY037, 2022JBQY007, and 2024YJS047, by the Beijing Natural Science Foundation under Grant L231019, by the National Natural Science Foundation of China under Grants 62276019 and 62306028, by the Shenzhen Science and Technology Program Project under Grant KJZD20240903102742055, and by the National Research Foundation Fellowship (NRF- NRFF13-2021-0006), Singapore.

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

# A. Theoretical Proofs

## A.1. Proof of Lemma (4.2)

Let $W_k$ be the parameters for task $k$, and let $T_i$ be the task vector of task $i$. The feature shift at layer $l$ caused by $T_i$ is defined as:

$$\Delta f_{k|i}^l = f_k^l(W_k + T_i) - f_k^l(W_k).$$

We aim to prove the following bound:

$$\|\Delta f_{k|i}^l\| \le \gamma_l \|\Delta f_{k|i}^{l-1}\| + \|\Delta \hat{f}_{k|i}^l\|.$$

where $\Delta f_{k|i}^{l-1} = f_k^{l-1}(W_k + T_i) - f_k^{l-1}(W_k)$, and $\Delta \hat{f}_{k|i}^l = \hat{f}_k^l(W_k + T_i) - f_k^l(W_k)$.

*Proof.* The feature shift at layer $l$ can be decomposed as:

$$
\begin{aligned}
\Delta f_{k|i}^l &= f_k^l(f_k^{l-1}(W_k + T_i); W_k^l + T_i^l) - f_k^l(f_k^{l-1}(W_k); W_k^l) \\
&= f_k^l(f_k^{l-1}(W_k + T_i); W_k^l + T_i^l) - f_k^l(f_k^{l-1}(W_k); W_k^l + T_i^l) + f_k^l(f_k^{l-1}(W_k); W_k^l + T_i^l) - f_k^l(f_k^{l-1}(W_k); W_k^l).
\end{aligned}
\tag{18}
$$

Under the norm-induced triangular inequality, we have:

$$\|\Delta f_{k|i}^l\| \le \|f_k^l(f_k^{l-1}(W_k + T_i); W_k^l + T_i^l) - f_k^l(f_k^{l-1}(W_k); W_k^l + T_i^l)\| + \|f_k^l(f_k^{l-1}(W_k); W_k^l + T_i^l) - f_k^l(f_k^{l-1}(W_k); W_k^l)\|.
\tag{19}$$

- For the first part, we apply the $\gamma_l$-Lipschitz continuity of $f_k^l$:

$$\|f_k^l(f_k^{l-1}(W_k + T_i); W_k^l + T_i^l) - f_k^l(f_k^{l-1}(W_k); W_k^l + T_i^l)\| \le \gamma_l \|\Delta f_{k|i}^{l-1}\|.$$

- For the second part:

$$\|f_k^l(f_k^{l-1}(W_k); W_k^l + T_i^l) - f_k^l(f_k^{l-1}(W_k); W_k^l)\| = \|\Delta \hat{f}_{k|i}^l\|.$$

Combining these two terms, we conclude:

$$\|\Delta f_{k|i}^l\| \le \gamma_l \|\Delta f_{k|i}^{l-1}\| + \|\Delta \hat{f}_{k|i}^l\|.$$

$\square$

## A.2. Proof of Theorem 4.4

**Theorem 4.4** (An Upper Bound on Knowledge Conflict).

Suppose that Assumptions 4.1 and 4.3 hold for any task $k$ and task vector $T_i$, the knowledge conflict follows

$$|\Delta \mathcal{L}_{k|i}| \le \beta \sum_{l=1}^{L} \Big( \prod_{m=l+1}^{L} \gamma_m \Big) \|\Delta \hat{f}_{k|i}^l\|.
\tag{20}$$

*Proof.* Recall that in Lemma (4.2), we derive the following recursive Lipschitz bound:

$$\|\Delta f_{k|i}^l\| \le \gamma_l \|\Delta f_{k|i}^{l-1}\| + \|\Delta \hat{f}_{k|i}^l\|.$$

We use recursive substitution starting from layer $L$ down to $l = 1$:

$$
\begin{aligned}
\|\Delta f_{k|i}^L\| &\leq \gamma_L \|\Delta f_{k|i}^{L-1}\| + \|\Delta \hat{f}_{k|i}^L\| \\
&\leq \gamma_L \left( \gamma_{L-1} \|\Delta f_{k|i}^{L-2}\| + \|\Delta \hat{f}_{k|i}^{L-1}\| \right) + \|\Delta \hat{f}_{k|i}^L\| = \gamma_L \gamma_{L-1} \|\Delta f_{k|i}^{L-2}\| + \gamma_L \|\Delta \hat{f}_{k|i}^{L-1}\| + \|\Delta \hat{f}_{k|i}^L\| \\
&\leq \sum_{l=1}^{L} \left( \prod_{m=l+1}^{L} \gamma_m \right) \|\Delta \hat{f}_{k|i}^l\|.
\end{aligned}
\tag{21}
$$

Using Assumption 4.3, which states that $\mathcal{L}$ is $\beta$-Lipschitz continuous with respect to the network's final output, we conclude:

$$
|\Delta \mathcal{L}_{k|i}| \leq \beta \sum_{l=1}^{L} \left( \prod_{m=l+1}^{L} \gamma_m \right) \|\Delta \hat{f}_{k|i}^l\|.
\tag{22}
$$

$\square$

### A.3. Frobenius Norm Decomposition of Orthogonal Projection

Assume that $B$ has orthonormal columns (i.e., $B^T B = I$), then, for any matrix $M$, the following equation holds:

$$
\left\| MBB^\top \right\|_F^2 + \left\| M - MBB^\top \right\|_F^2 = \|M\|_F^2 .
\tag{23}
$$

*Proof.*

$$
\begin{aligned}
&\left\| MBB^\top \right\|_F^2 + \left\| M - MBB^\top \right\|_F^2 \\
=& \left\| MBB^\top \right\|_F^2 + \left\| M(I - BB^\top) \right\|_F^2 \\
=& Tr((MBB^\top)^\top MBB^\top) + Tr((M(I - BB^\top))^\top M(I - BB^\top)) \\
=& Tr(BB^\top M^\top MBB^\top) + Tr((I - BB^\top)^\top M^\top M(I - BB^\top)) \\
=& Tr(BB^\top BB^\top M^\top M) + Tr((I - BB^\top)(I - BB^\top)^\top M^\top M) \\
=& Tr(BB^\top M^\top M) + Tr((I - BB^\top)(I^\top - (BB^\top)^\top)M^\top M) \\
=& Tr(BB^\top M^\top M) + Tr((I - BB^\top)(I - BB^\top)M^\top M) \\
=& Tr(BB^\top M^\top M) + Tr((I - BB^\top - BB^\top + BB^\top BB^\top)M^\top M) \\
=& Tr(BB^\top M^\top M) + Tr((I - BB^\top)M^\top M) \\
=& Tr(M^\top M) \\
=& \|M\|_F^2
\end{aligned}
\tag{24}
$$

$\square$

### A.4. Solution of Eq.(9)

Let $X_k \in \mathbb{R}^{n \times d}$ be the feature and $T_i \in \mathbb{R}^{d \times h}$ represent the task vector of the task $i$. Now our target is learning a group of **removal** basis $B_k \in \mathbb{R}^{h \times c}$ for task $k$ such that:

$$
\max_{B_k} \sum_{i \neq k} \left\| X_k T_i B_k B_k^\top \right\|_F^2 - \lambda \left\| X_i T_i B_k B_k^\top \right\|_F^2
\tag{25}
$$

Then we have:

$$\sum_{i \neq k} \left\| X_k T_i B_k B_k^\top \right\|_F^2 - \lambda \left\| X_i T_i B_k B_k^\top \right\|_F^2$$

$$= \sum_{i \neq k} \mathrm{Tr}(X_k T_i B_k B_k^\top T_i^\top X_k^\top) - \lambda Tr(X_i T_i B_k B_k^\top T_i^\top X_i^\top)$$

$$= \sum_{i \neq k} \mathrm{Tr}(T_i B_k B_k^\top T_i^\top X_k^\top X_k) - \lambda Tr(T_i B_k B_k^\top T_i^\top X_i^\top X_i)$$

$$= \sum_{i \neq k} \mathrm{Tr}\left(T_i B_k B_k^\top T_i^\top \left(X_k^\top X_k - \lambda X_i^\top X_i\right)\right)$$

$$= \sum_{i \neq k} \mathrm{Tr}\left(B_k B_k^\top T_i^\top \left(X_k^\top X_k - \lambda X_i^\top X_i\right) T_i\right) \qquad (26)$$

$$= \mathrm{Tr}\left(B_k B_k^\top \sum_{i \neq k} T_i^\top \left(X_k^\top X_k - \lambda X_i^\top X_i\right) T_i\right)$$

$$= \mathrm{Tr}\left(B_k^\top \underbrace{\left(\sum_{i \neq k} T_i^\top \left(X_k^\top X_k - \lambda X_i^\top X_i\right) T_i\right)}_{G} B_k\right)$$

The above equation implies that the largest $c$ eigenvectors of $G$ admit an optimal solution.

### A.5. Solution of Eq.(12)

Let $x_k \in \mathbb{R}^d$ be the feature and $T_i \in \mathbb{R}^d$ represents the task vector of the task $i$ for a scaler of a normalization layer. Now our target is learning a group of **removal** binary mask $m_k \in \{0, 1\}^d$ for task $k$ such that:

$$\max_{m_k} \sum_{i \neq k} \left( \sum_{x_k^l} \left\| x_k^l \circ T_i^l \circ m_k^l \right\|^2 - \lambda \sum_{x_i^l} \left\| x_i^l \circ T_i^l \circ m_k^l \right\|^2 \right). \qquad (27)$$

Then we have:

$$\sum_{i \neq k} \left( \sum_{x_k^l} \left\| x_k^l \circ T_i^l \circ m_k^l \right\|^2 - \lambda \sum_{x_i^l} \left\| x_i^l \circ T_i^l \circ m_k^l \right\|^2 \right)$$

$$= \sum_{i \neq k} \left( \sum_{x_k} \sum_{z=1}^d (x_{k,z} T_{i,z} m_{k,z})^2 - \lambda \sum_{x_i} \sum_{z=1}^d (x_{i,z} T_{i,z} m_{k,z})^2 \right)$$

$$= \sum_{z=1}^d m_{k,z}^2 \sum_{i \neq k} \left( \sum_{x_k} (x_{k,z} T_{i,z})^2 - \lambda \sum_{x_i} (x_{i,z} T_{i,z})^2 \right) \qquad (28)$$

$$= \sum_{z=1}^d m_{k,z} \underbrace{\sum_{i \neq k} \left( \sum_{x_k} (x_{k,z} T_{i,z})^2 - \lambda \sum_{x_i} (x_{i,z} T_{i,z})^2 \right)}_{g_z}$$

$$= \sum_{z=1}^d m_{i,k} g_z.$$

The above equation implies that in $m_k$, dimensions corresponding to the largest $c$ values of $g_z$ should be set to 1, while others should be 0.

### A.6. Solution of Eq.(13)

Let $x_k \in \mathbb{R}^d$ be the feature and $T_i \in \mathbb{R}^d$ represents the task vector of the task $i$ for shift parameters. Now our target is learning a group of **removal** binary mask $m_k \in \{0,1\}^d$ for task $k$ such that:

$$\max_{m_k} \sum_{i \neq k} \left( \sum_{x_k} \|T_i \circ m_k\|^2 - \lambda \sum_{x_i} \|T_i \circ m_k\|^2 \right). \tag{29}$$

Suppose $0 < \lambda < 1$ and all tasks have an equal amount of data $n$, then we have:

$$
\begin{aligned}
& \sum_{i \neq k} \left( \sum_{x_k} \|T_i \circ m_k\|^2 - \lambda \sum_{x_i} \|T_i \circ m_k\|^2 \right) \\
=& n \sum_{i \neq k} \left( \|T_i \circ m_k\|^2 - \lambda \|T_i \circ m_k\|^2 \right) \\
=& n(1-\lambda) \sum_{i \neq k} \left( \|T_i \circ m_k\|^2 \right) \\
=& \sum_{i \neq k} \left( \sum_{x_k} \sum_{z=1}^d (x_{k,z} T_{i,z} m_{k,z})^2 - \lambda \sum_{x_i} \sum_{z=1}^d (x_{i,z} T_{i,z} m_{k,z})^2 \right) \\
=& n(1-\lambda) \sum_{i \neq k} \left( \sum_{z=1}^d T_{i,z}^2 m_{k,z}^2 \right) \\
=& n(1-\lambda) \sum_{z=1}^d m_{k,z} \underbrace{\sum_{i \neq k} T_{i,z}^2}_{g_z} \\
=& n(1-\lambda) \sum_{z=1}^d m_{i,k} g_z,
\end{aligned}
\tag{30}
$$

The above equation implies that in $m_k$, dimensions corresponding to the largest $c$ values of $g_z$ should be set to 1, while others should be 0.

## B. Experiment Details

This section introduces some additional details of experiments, including the detailes of the experimental environment, datasets, and baselines.

### B.1. Environment

All experiments detailed in our manuscript and appendix were conducted on a workstation running Ubuntu 16.04, equipped with 2 Intel Xeon 2.60GHz CPUs, 256 GB of memory, and 6 NVIDIA RTX3090 GPUs. We leverage Python 3.8 to implement all the methods.

### B.2. Datasets

Our experiments strictly follow Task Arithmetic (Ilharco et al., 2023b) and leverage the following eight widely-used image classification datasets:

- **SUN397** (Xiao et al., 2016): A large-scale scene classification dataset containing 108,754 images organized into 397 categories. Each category includes at least 100 images, making it a diverse benchmark for scene recognition tasks.

- **Stanford Cars (Cars)** (Krause et al., 2013): A fine-grained car classification dataset featuring 16,185 images of 196 distinct car models. The dataset is evenly divided into training and test splits, enabling robust model evaluation.

- **RESISC45** (Cheng et al., 2017): A remote sensing image classification dataset with 31,500 images spanning 45 scene classes. Each class contains approximately 700 images, covering a variety of landscapes and man-made structures.

- **EuroSAT** (Helber et al., 2019): A satellite imagery dataset designed for land-use and land-cover classification, consisting of 27,000 labeled and geo-referenced images distributed among 10 categories, such as forests, residential areas, and agricultural fields.

- **SVHN** (Netzer et al., 2011): A digit classification dataset derived from real-world house numbers captured in Google Street View. It includes 10 digit classes, with 73,257 training images, 26,032 test images, and an additional 531,131 samples for extended training.

- **GTSRB** (Stallkamp et al., 2011): The German Traffic Sign Recognition Benchmark, comprising over 50,000 images across 43 traffic sign categories. This dataset is widely used for evaluating traffic sign recognition systems.

- **MNIST** (LeCun & Cortes, 2010): A classic handwritten digit classification dataset containing 60,000 training images and 10,000 test images, evenly distributed across 10 digit classes.

- **DTD** (Cimpoi et al., 2014): The Describable Textures Dataset, which includes 5,640 images spanning 47 texture categories, with around 120 images per category. It is designed for texture recognition tasks.

We also leverage the following six vision-language datasets:

- **COCO Caption** (Chen et al., 2015): A large-scale image captioning dataset derived from the MS COCO dataset. It contains over 330,000 images, with each image annotated with five different captions, facilitating training for generating natural language descriptions of images.

- **Flickr30k Caption** (Plummer et al., 2015): A dataset for image captioning and retrieval tasks, consisting of 31,000 images sourced from Flickr. Each image is paired with five descriptive sentences, capturing a variety of objects, scenes, and actions in the images.

- **TextCaps** (Sidorov et al., 2020): A challenging image captioning dataset focusing on reasoning over both visual and textual content in images. It includes 145,000 image-caption pairs, where captions must integrate text from the image to provide meaningful descriptions.

- **OKVQA** (Marino et al., 2019): A knowledge-based visual question-answering dataset designed to evaluate the ability to answer open-ended questions about images using external knowledge. It consists of more than 14,000 questions and corresponding answers requiring reasoning beyond the image content.

- **TextVQA** (Singh et al., 2019): A dataset for visual question answering where reading and understanding text present in images is crucial. It includes over 45,336 questions across 28,408 images, requiring the integration of textual and visual reasoning to generate accurate answers.

- **ScienceQA** (Lu et al., 2022): A multi-modal dataset designed for science-related question answering. It contains over 21,208 multi-modal multiple-choice questions paired with textual explanations and images across various scientific disciplines, such as biology, physics, and chemistry, supporting reasoning-based evaluation.

### B.3. Baselines.

Our experiments are associated with several baseline approaches. The details of these baselines are as follows:

- **Pretrained** directly employs a pretrained model to predict across multiple tasks. Since it does not incorporate any downstream task-specific information during model training, its performance on downstream tasks is typically suboptimal.

- **Individual**. In this approach, an independent finetuned model is used for each task. While it avoids interference between tasks, it cannot perform multiple tasks simultaneously. It serves as a reference *upper bound* for model merging approaches.

- **Traditional MTL** aggregates the original training data from all tasks to train a single multi-task model.

- **Weight Averaging** directly averages model parameters from multiple tasks into a single model, enabling multi-task learning without additional training.

- **Fisher Merging** (Matena & Raffel, 2022) leverages the Fisher information matrix to assess parameter importance, merging model parameters based on this importance.

- **RegMean** (Jin et al., 2023) refines weight matrices by adjusting and linearly combining rows, utilizing statistical information derived from the training data.

- **Task Arithmetic** (Ilharco et al., 2023b) introduces the concept of a "task vector," defined as the difference between finetuned model parameters and pretrained model parameters. Multiple task vectors are then combined and added to the pretrained model to facilitate multi-task learning.

- **Ties-Merging** (Yadav et al., 2023) eliminates unimportant parameters from the task vector and resolves sign conflicts among parameters, reducing interference during the final task vector merging process.

- **TATR** (Sun et al., 2025). This method advances task arithmetic by restricting the merging within a trust region to mitigate knowledge conflict.

- **TATR & Ties-Merging** (Sun et al., 2025; Yadav et al., 2023). This method combines the trust region restriction in TATR into Ties-Merging to enhance the performance.

- **Consensus Merging (Wang et al., 2024)** computing a group of masks for each task vector to minimize the distance in parameter space between the merged model and the finetuned model.

- **PCB Merging (Du et al., 2024)** trims components in the task vector that have small magnitudes and are not significantly related to other tasks.

## C. Additional Experiments

### C.1. Comparison on ViT-B/16

Table 6 presents the results of various model merging methods using the ViT-B/16 architecture. As we can see, CAT Merging significantly improves the multi-task performance of Task Arithmetic, raising the average performance from 73.8% to 82.1%.

*Table 6.* Multi-task performance when merging ViT-B/16 models on eight tasks.

| Method | SUN397 | Cars | RESISC45 | EuroSAT | SVHN | GTSRB | MNIST | DTD | Avg Acc |
|---|---|---|---|---|---|---|---|---|---|
| Pretrained | 63.8 | 64.6 | 65.7 | 54.5 | 52.0 | 43.3 | 51.7 | 45.1 | 55.0 |
| Individual | 81.8 | 86.8 | 96.9 | 99.7 | 97.8 | 99.1 | 99.7 | 82.0 | 92.9 |
| Weight Averaging | 67.7 | 70.0 | 75.3 | 79.5 | 74.9 | 60.1 | 94.4 | 43.8 | 70.7 |
| Fisher Merging | 68.5 | 69.9 | 75.2 | 80.4 | 73.2 | 61.2 | 94.5 | 50.7 | 71.7 |
| RegMean | 69.1 | 71.6 | 77.6 | 88.8 | 83.7 | 70.2 | 96.9 | 54.6 | 76.6 |
| Task Arithmetic | 61.1 | 65.9 | 74.0 | 76.2 | 88.0 | 73.9 | 98.4 | 53.0 | 73.8 |
| Ties-Merging | 69.1 | 72.5 | 80.5 | 84.0 | 85.0 | 71.5 | 98.1 | 54.9 | 77.0 |
| TATR | 67.4 | 70.4 | 77.9 | 81.7 | 87.6 | 77.2 | 98.3 | 55.6 | 77.0 |
| CAT Merging (ours) | 72.9 | 75.9 | 83.1 | 92.8 | 88.2 | 82.7 | 98.8 | 62.7 | 82.1 |

### C.2. Comparing with Test-time Training based Methods

Table 7 shows that our CAT Merging achieves comparable or superior performance relative to two representative training-based techniques, demonstrating its effectiveness without incurring additional computational costs.

*Table 7.* Multi-task performance when merging on eight vision tasks.

| Method | SUN397 | Cars | RESISC45 | EuroSAT | SVHN | GTSRB | MNIST | DTD | Avg Acc |
|---|---|---|---|---|---|---|---|---|---|
| | | | | *ViT-B/32* | | | | | |
| TW AdaMerging | 58.0 | 53.2 | 68.8 | 85.7 | 81.1 | 84.4 | 92.4 | 44.8 | 71.1 |
| TW AdaMerging++ | 60.8 | 56.9 | 73.1 | 83.4 | 87.3 | 82.4 | 95.7 | 50.1 | 73.7 |
| LW AdaMerging | 64.5 | 68.1 | 79.2 | 93.8 | 87.0 | 91.9 | 97.5 | 59.1 | 80.1 |
| LW AdaMerging++ | 66.6 | 68.3 | 82.2 | 94.2 | 89.6 | 89.0 | 98.3 | 60.6 | 81.1 |
| Surgery Merging | 63.8 | 59.9 | 83.3 | 97.9 | 87.0 | 87.0 | 98.6 | 69.4 | 80.9 |
| CAT Merging (ours) | 68.1 | 65.4 | 80.5 | 89.5 | 85.5 | 78.5 | 98.6 | 60.7 | 78.3 |
| | | | | *ViT-L/14* | | | | | |
| AdaMerging | 79.0 | 90.3 | 90.8 | 96.2 | 93.4 | 98.0 | 99.0 | 79.9 | 90.8 |
| AdaMerging++ | 79.4 | 90.3 | 91.6 | 97.4 | 93.4 | 97.5 | 99.0 | 79.2 | 91.0 |
| Surgery Merging | 75.7 | 84.4 | 93.1 | 98.8 | 91.3 | 93.4 | 99.1 | 76.1 | 89.0 |
| CAT Merging (ours) | 78.7 | 88.5 | 91.1 | 96.3 | 91.3 | 95.7 | 99.4 | 75.7 | 89.6 |

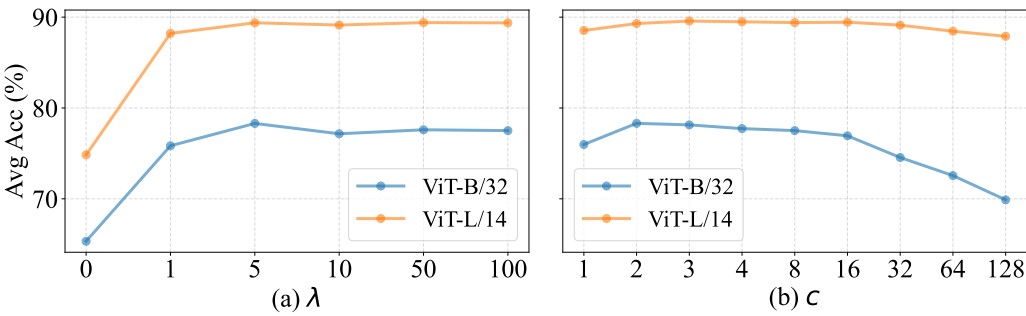

*Figure 4.* Average accuracy (%) of CAT Merging on eight vision tasks with different values of $\lambda$ (a) and $c$ (b).

## C.3. Generalization Comparison

This section explores the generalization ability of CAT Merging. Specifically, we merge models using task vectors from six tasks and evaluate their performance on two unseen tasks (MNIST and EuroSAT). The results in Table 8 show that CAT Merging outperforms all baselines on both seen and unseen datasets, with an average performance improvement of 3.4% and 2.6%, respectively. This improvement in generalization is attributed to CAT Merging's ability to handle knowledge conflicts, ensuring that model updates move toward a more globally optimal direction.

*Table 8.* Generalization results on two unseen tasks when merging ViT-B/32 models on six tasks.

| Method | SUN397 | Cars | RESISC45 | DTD | SVHN | GTSRB | Avg Acc | MNIST | EuroSAT | Avg Acc |
|---|---|---|---|---|---|---|---|---|---|---|
| Task Arithmetic | 63.3 | 62.4 | 75.1 | 57.8 | 84.6 | 80.4 | 70.6 | 77.2 | 46.2 | 61.7 |
| Ties-Merging | 67.8 | 66.2 | 77.2 | 56.7 | 77.1 | 70.9 | 69.3 | 75.9 | 43.3 | 59.6 |
| TATR | 66.0 | 64.1 | 77.9 | 60.1 | 83.9 | 81.8 | 72.3 | 77.2 | 47.7 | 62.5 |
| CAT Merging (Ours) | 70.4 | 68.4 | 85.3 | 63.6 | 82.8 | 83.8 | **75.7** | 77.8 | 52.3 | **65.1** |

## C.4. Sensitivity Analysis

This section analyzes the sensitivity of two additional hyper-parameters $\lambda$ and $c$.

**Sensitivity analysis of weight $\lambda$.** In Figure 4 (a), $\lambda$ significantly impacts performance only when set to 0, where the accuracy drops sharply to 65.33% (ViT-B/32) and 74.84% (ViT-L/14). This indicates that neglecting task-specific knowledge severely degrades the results. For $\lambda > 0$, the performance remains stable across a wide range of values, demonstrating the robustness of CAT Merging.

**Sensitivity analysis of $c$.** $c$ affects the number of task vector components are trimmed. As illustrated in Figure 4 (b), the performance of CAT Merging remains stable for small $c$, peaking at $c = 2$ for ViT-B/32 and $c = 3$ for ViT-L/14. Larger values lead to a partial decline as more knowledge is discarded, with ViT-B/32 dropping to 69.89% at $c = 128$.

*Table 9.* Computational complexity comparison (in seconds) for merging ViT-B/32 and ViT-L/14 models across eight vision tasks, measured on a single RTX 3090 GPU.

| Method | TA w/ Surgery | AdaMerging | TATR | PCB Merging | CAT Merging (ours) |
|---|---|---|---|---|---|
| ViT-B/32 | 12621 | 8276 | 176 | 43 | 46 |
| ViT-L/14 | 36826 | 16299 | 283 | 131 | 150 |

### C.5. Analysis of Computational Complexity

The computational overhead of CAT Merging is reasonable and practically efficient. Specifically, CAT Merging involves two main steps:

- **Feature Extraction:** This step is lightweight and efficient, requiring only a small number (2–3 per task) of unlabeled samples;

- **Eigendecomposition:** While eigendecomposition has theoretically higher computational complexity, in practice, we efficiently mitigate this through GPU parallelization. Moreover, CAT Merging only requires the eigenvectors corresponding to the top-$c$ (2-4 in our work) eigenvalues, enabling further acceleration through specialized methods (e.g., torch.lobpcg). Empirical results (provided in the table 9, measured on a single RTX3090 GPU in seconds) demonstrate that CAT Merging significantly outperforms training-based counterparts (e.g., TA w/ Surgery (Yang et al., 2024a), AdaMerging (Yang et al., 2024b)) in terms of computational efficiency.

*Table 10.* Balance comparison for merging ViT-B/32 and ViT-L/14 models across eight vision tasks.

| Method | Fisher Merging | RegMean | Task Arithmetic | PCB Merging | CAT Merging (ours) |
|---|---|---|---|---|---|
| ViT-B/32 | 13.78 | 8.46 | 8.95 | 6.85 | 6.21 |
| ViT-L/14 | 6.79 | 6.86 | 5.11 | 3.49 | 2.51 |

### C.6. Analysis of the Balance during Merging

As shown in Table 1 and 2, while CAT Merging achieves superior average performance, it does not always yield the highest accuracy on every individual dataset.

Specifically, we observe that Fisher Merging exhibits better performance on certain datasets (e.g., Cars and SUN397), likely because its weighting mechanism, based on the Fisher information matrix, implicitly prioritizes tasks with weaker performance (larger gradients produce higher Fisher information scores). Conversely, PCB Merging achieves superior performance on datasets like SVHN and MNIST, where masking low-magnitude vector components implicitly favors tasks with stronger finetuning outcomes (assuming larger vector magnitudes correlate with greater task specialization).

However, both Fisher and PCB merging tend to perform less consistently across other tasks. In contrast, our CAT Merging framework explicitly targets inter-task knowledge conflicts and aims for a balanced integration across tasks. To quantitatively illustrate this balance, we measured the standard deviation of accuracy drops (defined as the accuracy difference between task-specific models and the merged model) across tasks. As shown in the table 10, CAT Merging demonstrates significantly lower variance, reflecting its ability to merge multiple tasks more evenly, without undue preference towards any particular task.

### C.7. Analysis of Knowledge Conflict

This section provides more evidence about the effective mitigation of knowledge conflict in CAT Merging. Figure 5 visualizes the knowledge conflict during merging under different merging coefficients. As can be seen, CAT Merging consistently has lesser knowledge conflict than Task Arithmetic.

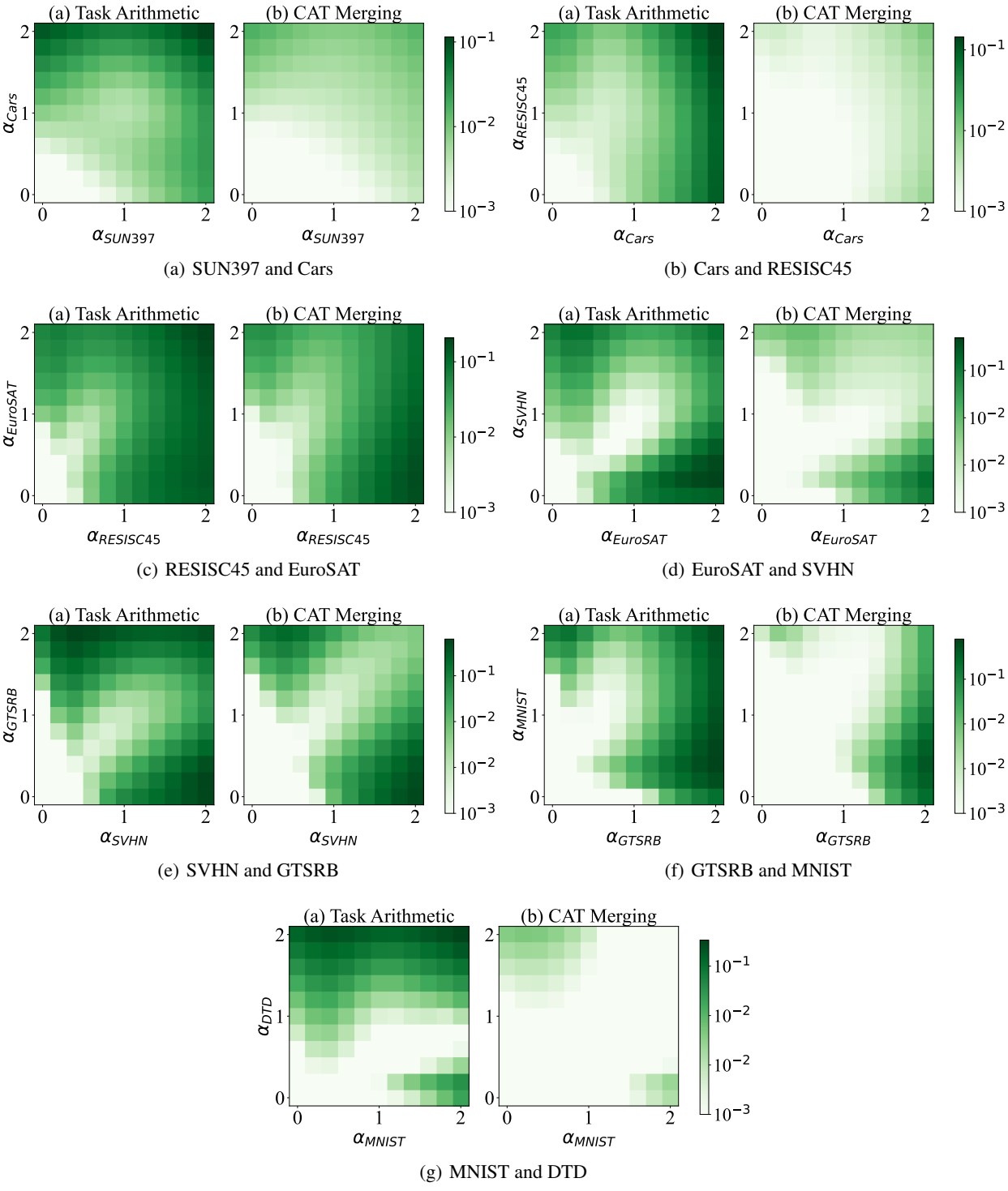

*Figure 5.* Visualization of knowledge conflict when merging two ViT-L/14 models.

