# OpenReview forum: "CAT Merging: A Training-Free Approach for Resolving Conflicts in Model Merging"
_ICML.cc/2025/Conference — ICML 2025 poster_

### Official Review · Reviewer_oc91 · 2025-03-10

**Overall Recommendation:** 3

**Summary:**

## Summary

The paper introduces CAT Merging, a training-free framework for merging multiple expert models while mitigating knowledge conflicts. Existing methods, such as task vectors, merge models by accumulating task vector weights, but conflicting components across tasks can lead to performance degradation. CAT Merging addresses this by formulating conflict resolution as an optimization problem to trim conflict-prone components. Experimental results show that CAT Merging significantly reduces knowledge conflicts and show (in most cases) accuracy gains of a few percentages over SoTA.

## Update after rebuttal
I appreciate the response by the authors. I maintain my score.

**Claims And Evidence:**

Yes. I found the paper very clear in this respect. The presented claims are well supported by theoretical arguments and experiments.

**Essential References Not Discussed:**

The references are sufficient.

**Experimental Designs Or Analyses:**

The experimental results seem sound, but the reviewer is concerned that the performance gains are relatively modest. Given that the proposed method does not achieve the best performance across all tested settings despite its solid theoretical foundation, it would be valuable to analyze and discuss the sources of suboptimality.

**Methods And Evaluation Criteria:**

Yes, the method is thoroughly evaluated across multiple architectures and datasets. Additionally, the paper includes a well-conducted ablation study and sensitivity analysis.

**Other Comments Or Suggestions:**

In Sec. 5.1, it would be helpful to clarify the significance of the removal basis B. It is also unclear if B is a binary matrix. Explicitly stating this would improve clarity.

**Other Strengths And Weaknesses:**

The paper presents a theoretically grounded approach to conflict resolution in task vectors, making it an interesting read. However, the somewhat low score is primarily due to the method's weak performance compared to the baselines and the lack of clarity on the underlying reasons for this suboptimality.

**Questions For Authors:**

What factors contribute to the suboptimal performance of CAT Merging?

**Relation To Broader Scientific Literature:**

The authors provide a clear presentation of their method and its contributions to the scientific literature. I would like to bring their attention to the following concurrent work: https://openreview.net/forum?id=4wuvmJRAU4.

**Theoretical Claims:**

I checked the correctness of the claims in the main paper. I didn't look into the proofs provided in the appendix.

---

> ### Author Rebuttal · Authors · 2025-04-01
>
> **Q4.1: Pay attention to the concurrent work.**
>
> **A4.1:** Thank you for highlighting the concurrent work, "Interfering with Interference: Blind Shuffling and Superposition for Better Multi-Model Compression," which addresses interference during multi-model merging through random layer shuffling and orthogonal transformations.
>
> We appreciate the relevance, as both this concurrent paper and our CAT Merging framework aim to mitigate interference between task vectors. However, there are fundamental differences between their approach and ours:
>
> **Motivation and Insight**: The concurrent work attributes interference primarily to task vector similarity and proposes randomization-based techniques (layer shuffling and task vector superposition) to increase orthogonality. In contrast, our work explicitly identifies conflict-prone components within task vectors and resolves them through targeted, parameter-specific trimming strategies.
>
> **Methodological Differences**: Their approach depends on randomized transformations, necessitating task-specific decoding (inverse transformations) at inference time. This requirement limits applicability in scenarios involving mixed-task batches. In contrast, CAT Merging enables seamless multi-task inference without task-specific decoding or per-sample routing, making it more suitable for practical deployments involving shared-task batches.
>
> We will include a discussion of this concurrent study in the related work section of our revised manuscript to position our contributions relative to this relevant work.
>
> **Q4.2: What factors lead to the suboptimal performance of CAT Merging?**
>
> **A4.2:** We thank the reviewer for highlighting this important point. While CAT Merging achieves superior **average** performance—improving accuracy by 2.5% (ViT-B/32) and 2.0% (ViT-L/14) compared to state-of-the-art methods—it is true that it does not always yield the highest accuracy on every individual dataset.
>
> Specifically, we observe that Fisher Merging exhibits better performance on certain datasets (e.g., Cars and SUN397), likely because its weighting mechanism, based on the Fisher information matrix, implicitly prioritizes tasks with weaker performance (larger gradients produce higher Fisher information scores). Conversely, PCB Merging achieves superior performance on datasets like SVHN and MNIST, where masking low-magnitude vector components implicitly favors tasks with stronger fine-tuning outcomes (assuming larger vector magnitudes correlate with greater task specialization).
>
> However, both Fisher and PCB merging tend to perform less consistently across other tasks. In contrast, our CAT Merging framework explicitly targets inter-task knowledge conflicts and aims for a balanced integration across tasks. To illustrate this balance quantitatively, we measured the standard deviation of accuracy drops (defined as the accuracy difference between task-specific models and the merged model) across tasks in the following table. It shows that CAT Merging demonstrates significantly lower variance, reflecting its ability to merge multiple tasks more evenly, without undue preference towards any particular task. We will clarify this analysis and discussion in the revised manuscript.
>
> |  | Fisher Merging | RegMean | Task Arithmetic | PCB Merging | CAT Merging (ours) |
> | --- | --- | --- | --- | --- | --- |
> | ViT-B/32 | 13.78 | 8.46 | 8.95 | 6.85 | **6.21** |
> | ViT-L/14 | 6.79 | 6.86 | 5.11 | 3.49 | **2.51** |
>
> **Q4.3: What is the significance of the removal basis? Is it binary?**
>
> **A4.3:** Thank you for this helpful suggestion. We will clarify this point explicitly in the revised manuscript. Specifically, the removal basis $B$ is a real-valued matrix (not binary). It is optimized to define a task-specific subspace, within which conflict-prone components are identified and suppressed via orthogonal projection. The basis matrix $B$ thus plays a crucial role in effectively mitigating knowledge conflicts during the merging process.

---

### Official Review · Reviewer_jxyf · 2025-03-14

**Overall Recommendation:** 3

**Summary:**

The paper introduces Conflict-Aware Task Merging, a training-free model merging method that addresses knowledge conflicts in multi-task model merging. The meaning of knowledge conflicts is that existing methods, such as Task Arithmetic, suffer from conflicts when integrating multiple fine-tuned task vectors, often resulting in performance degradation. CAT Merging mitigates these conflicts by selectively trimming conflict-prone components from task vectors based on different parameter types: Feature projection for linear weights, Masking for scaling parameters in normalization layers and Masking for shifting parameters. The method is evaluated on vision and vision-language tasks, demonstrating up to 4.7% accuracy improvement on ViT-B/32 and 2.0% on ViT-L/14 compared to state-of-the-art model merging techniques.

**Claims And Evidence:**

S1: The paper is well-written and clearly organized, with detailed algorithmic descriptions (e.g., Algorithm 1) and thorough theoretical derivations (e.g., Theorem 5.1), making it easy for readers to understand the problem, methodology, and experiments. Additionally, the related work section is logically organized and highly readable;

**Essential References Not Discussed:**

N.A.

**Experimental Designs Or Analyses:**

W1: The paper provides an analysis of previous methods, such as Fisher Merging and Ties-Merging, which are discussed in sections like “Knowledge Conflict” and “Introduction”. However, the sentence in the Introduction --- “As the example shown in the figure 1, magnitude-based methods such as Ties-Merging prioritize task vectors with larger magnitudes (e.g., Task 1) while trimming dimensions with smaller magnitudes, inadvertently discarding critical information from Task 2.” may require further clarification. In contrast, the statement in the Knowledge Conflict section --- “Most existing methods such as Fisher Merging or Ties Merging, implicitly prioritize preserving intra-task knowledge while neglecting the inter-task knowledge conflict” is more direct and could be placed earlier in the paper to better introduce the problem.

W6: The experimental evaluation does not include studies on large language models (LLM), which may limit the understanding of the method’s applicability in broader contexts.

W8: The paper evaluates CAT Merging on ViT and BLIP models, but does not test its applicability to LLMs.

W10: While training-based methods have additional costs, it would be useful to see a comparison against test-time adapted merging techniques to better contextualize the trade-offs.

**Methods And Evaluation Criteria:**

S3: The method is sound, and the idea of modeling the knowledge conflict through inter-task knowledge conflict and intra-task knowledge deviation is good. Additionally, the proposal and the design of Φ_k are well-designed;

**Other Comments Or Suggestions:**

W2: The formatting of Equation (7) appears to have an issue, as there seems to be an extra Δ symbol.

W3: There seems to be an issue with Equation (15), where it likely misses parentheses (( and )). Additionally, in the term ∥T_i^l+T_i^l∘m_k^l ∥^2, there should be a - instead of a +.

W5: Additional comparisons on inference speed and computational overhead during the merging process would offer a more complete evaluation of the method’s efficiency.

**Other Strengths And Weaknesses:**

The writing is clear, and the logic is well-structured. The method is novel and sound, and I appreciate the modeling of knowledge conflict, the proposal of the Φ_k operation, and the design of the parameter-specific strategies. Additionally, I think the method is reproducible. Considering the strengths and weaknesses discussed above, I find this paper to be a solid contribution. I also hope that the work presented in this paper will see broader applications in real-world settings.

**Questions For Authors:**

Please refer to the above comments.

**Relation To Broader Scientific Literature:**

S5：The approach is training-free and only relies on a lightweight forward pass with a small number of exemplars.

**Theoretical Claims:**

S2: The mathematical derivations in the paper are well executed, with sufficient details provided throughout. Additionally, the proofs included in the appendix are clear and well-structured, making it easy for readers to follow and understand;

S4: The approach decomposes the global optimization problem into layer-wise sub-problems, which not only simplifies the complex merging process but also provides valuable theoretical insights.

W7: The method’s reliance on theoretical assumptions (e.g., Lipschitz continuity) might not fully capture the complexities encountered in real-world scenarios; a discussion on potential limitations would be valuable.

W9: While projection for linear weights and masking for normalization/shift parameters seem reasonable, the paper does not provide a principled justification for why these are the best strategies.

---

> ### Author Rebuttal · Authors · 2025-04-01
>
> **Q3.1: Writing issues (W1-4).**
>
> **A3.1:**  Thanks for the suggestions. We will revise them and thoroughly double-check the manuscript to avoid similar issues.
>
> **Q3.2: Comparisons on inference speed and computational overhead (W5).**
>
> **A3.2:**
>
> **The inference speed** remains consistent with that of the individual models, as CAT Merging produces a unified model to conduct inference without adding extra layers.
>
> **The computational overhead** of CAT Merging is reasonable and practically efficient. Specifically, CAT Merging involves two main steps:
>
> 1. **Feature Extraction:** This step is lightweight and efficient, requiring only a small number (2–3 per task) of unlabeled samples.
> 2. **Eigendecomposition:** While eigendecomposition has theoretically higher computational complexity, in practice, we efficiently mitigate this through GPU parallelization. Moreover, CAT Merging only requires the eigenvectors corresponding to the top-*c* (2-4 in our work) eigenvalues, enabling further acceleration through specialized methods (e.g., `torch.lobpcg`). Empirical results (provided in the table below, measured on a single RTX3090 GPU in seconds) demonstrate that CAT Merging significantly outperforms training-based counterparts (e.g., TA w/ Surgery, AdaMerging) in terms of computational efficiency.
>
> |  | ViT-B/32 | ViT-L/14 |
> | --- | --- | --- |
> | PCB Merging | 43 | 131 |
> | CAT Merging (ours) | 46 | 150 |
> | TATR | 176 | 283 |
> | TA w/ Surgery | 12621 | 36826 |
> | AdaMerging | 8276 | 16299 |
>
> **Q3.3: Comparisons of LLMs (W6 and W8).**
>
> **A3.3:**  Thanks for your suggestion.
>
> To further validate CAT Merging on language tasks, we conducted additional experiments using RoBERTa as the backbone model on the GLUE benchmark, which comprises eight diverse NLP tasks, including classification and regression (STS-B). We report accuracy for classification tasks and the mean of Pearson and Spearman correlations for the regression task. As summarized in the table below, CAT Merging consistently achieves superior average performance compared to existing state-of-the-art merging methods, demonstrating its effective generalization and robustness in language model merging scenarios.
>
> | **Algorithm** | **cola** | **mnli** | **mrpc** | **qnli** | **qqp** | **rte** | **sst2** | **stsb** | **Average** | #best |
> | --- | --- | --- | --- | --- | --- | --- | --- | --- | --- | --- |
> | Task Arithmetic | 6.68 | 66.23 | **78.46** | 78.62 | 72.69 | 53.43 | 83.49 | 27.10 | 58.34 | 1 |
> | Ties-Merging | 9.46 | 59.34 | 74.71 | 65.93 | 41.29 | 47.29 | 72.13 | 9.210 | 47.42 | 0 |
> | PCB Merging | 11.40 | 50.85 | 77.63 | 78.22 | 55.78 | 60.29 | 75.57 | **67.01** | 59.59 | 1 |
> | CAT Merging (Ours) | **33.20** | **72.33** | 68.22 | **82.92** | **76.05** | **62.82** | **89.33** | 15.57 | **62.56** | **6** |
>
> **Q3.4: The theoretical assumptions do not capture the complexities of real-world scenarios (W7).**
>
> **A3.4:** We agree with the reviewers that theoretical assumptions, such as Lipschitz continuity, may not fully capture the complexities encountered in real-world scenarios. Nevertheless, we would like to emphasize that the primary role of Theorem 5.1 is to provide theoretical insight and motivation for our layer-by-layer trimming strategy, rather than deriving tight practical bounds. For further details, please see our response A2.6. In the final version of our manuscript, we will clarify this point further and include an explicit discussion on the potential limitations of our method.
>
>
> **Q3.5: Are the projection and masking the best strategies (W9)?**
>
> **A3.5:**  Good question. Our proposed strategies—projection for linear weights and masking for normalization parameters—are empirically motivated heuristics chosen for their computational efficiency and practical effectiveness in mitigating parameter conflicts. While these methods achieve strong empirical results, we acknowledge they are not theoretically guaranteed to be optimal. In future work, we aim to explore principled approaches, such as weighted averaging or Mixture-of-Experts (MoE), to further refine and theoretically ground our conflict mitigation techniques.
>
> **Q3.6: Comparing with training-based methods (W10).**
>
> **A3.6:**  Thank you for this suggestion. The following table shows that our CAT Merging achieves comparable or superior performance relative to two representative training-based techniques, demonstrating its effectiveness without incurring additional computational costs.
> |  | ViT-B/32 | ViT-L/14 |
> | --- | --- | --- |
> | TA w/ Surgery | 80.9 | 89.0 |
> | AdaMerging | 81.1 | 91.0 |
> | CAT Merging (ours) | 78.3 | 89.6 |

---

### Official Review · Reviewer_jpFU · 2025-03-14

**Overall Recommendation:** 2

**Summary:**

The paper proposes a novel model training-free model merging algorithm that removes the conflicting components of task vectors. This is done in a round robin fashion; for each task vector, the conflicting components of each other task vector are computed and removed from them. This is done with a projection for linear layers and masking for others. The method requires task-specific data, but authors show that even 1 sample per dataset yields good performance. The paper includes the standard 8-task CLIP benchmark proposed by the original task arithmetic paper as well as a vision-language benchmark.

**Claims And Evidence:**

The claims made in the submission are supported by empirical evidence. They would be stronger if LLM benchmarks would be included like in most model merging works.

**Essential References Not Discussed:**

I think all essential references are discussed.

**Experimental Designs Or Analyses:**

The experiments are sound and valid, following the standard practices of the field.

**Methods And Evaluation Criteria:**

The proposed methods and evaluation criteria do make sense. The authors should include all baselines from Tables 2 and 3 in their final experiment presented in Table 4 for completeness.

**Other Comments Or Suggestions:**

* L86 (2nd col) fix Ilharco citation with `\citet`
* L159 fix Guillermo et al.
* L159 “perfect weight disentanglement” is not a proper term, since weight disentanglement has not been introduced at all at this stage
* Include baselines in Figure 2. For figure 2a only present one model because the difference in performance between the two models might make the curves look flatter than they are.

**Other Strengths And Weaknesses:**

## Strengths

* The paper provides an algorithm that can work with as few as 1 sample per task to remove the conflicting components in the task vectors. This is a major strength of the paper and should be highlighted more imo.
* The performance of the proposed method is strong across multiple benchmarks and the experimental validation includes multiple baselines.
* The analysis of knowledge conflict is illuminating.
* Afaik, the idea is original and the result significant. Similar methodologies, such as Ties are referenced in the paper.

## Weaknesses

1. The theoretical claims are not well supported \- see comment on “Theoretical Claims”.
2. The writing of the paper can be improved:
   1. Motivation is task vectors with differing magnitudes (see Figure 1). However, the benchmarks are about vision classification with CLIP where the norms are approximately the same for all tasks. Hence, the motivation should be modified to mention the important task directions
   2. The mathematical formulations are too verbose and not enough intuition is provided. For instance, in Section 5: no need to have the superscript $l$, it makes the notation cumbersome. Similarly eq9 does not need the sum
3. L149-150: assuming alpha=1 is clearly wrong, since the performance of model merging papers highlights the scaling as a very important parameter. It should not be glanced over for brevity
4. Theorem 5.1 isnt the upper bound vacuous?
5. The paper claims that the procedure is lightweight but does not offer actual runtimes for comparison.

**Questions For Authors:**

1. How are the exemplars selected in Figure 2?
2. What exactly do you mean by “shift” parameters? The biases?
3. Are attention layers treated as linear layers?

**Relation To Broader Scientific Literature:**

The paper reviews adequately the related work and provides multiple recent baselines in the experimental section.

**Theoretical Claims:**

There are some issues with the theoretical claims in the paper that I encourage the authors to fix and/or clarify. The authors present their method for various types of layers in subsections 5.1, 5.2 and 5.3. A derivation is given and the proofs are provided in the appendix. In all cases, the end results refer to “top $c$ eigenvalues or components”. However, in Appendices A.3, A.4, A.5 the “proof” says that this is **"implied"** and does not provide any reasoning about how to actually optimize for $c$. I believe that this makes the derivation too informal and undermines the paper.

Furthermore, the bound provided in Theorem 5.1 is vacuous and does not provide any insight.

---

> ### Author Rebuttal · Authors · 2025-04-01
>
> **Q2.1: Results on LLM.**
>
> **A2.1:** Thanks for your suggestion. We conducted additional experiments using RoBERTa as the backbone model on the GLUE benchmark. As summarized in A3.3 below, CAT Merging consistently achieves superior average performance compared to existing state-of-the-art merging methods, demonstrating its effective generalization and robustness in language model merging scenarios.
>
> **Q2.2: Should all baselines from Tables 2 and 3 be included in Table 4?**
>
> **A2.2:** Thank you for this suggestion. Table 4 is intended as an ablation study specifically designed to analyze individual components within our proposed method; therefore, including all baselines from Tables 2 and 3 may not align well with its purpose. Could you please clarify whether you instead suggest adding all baselines to Table 3? If so, we will gladly incorporate the additional baselines to ensure a comprehensive and thorough evaluation.
>
> **Q2.3: How do the proofs in Appendices A.3, A.4, A.5 “imply top c eigenvalues or components”, and how is c optimized?**
>
> **A2.3:** Thank you for pointing out this issue. Specifically, the parameter c is a practical, data-dependent hyperparameter chosen to balance performance stability and exemplar efficiency. Given a fixed c, the optimality of selecting the top c eigenvectors can be rigorously justified. For instance, Eq. (21) in A.3 leads to the following optimization form:
>   $Tr(B^\top GB)=\sum_d \lambda_d ||B^\top v_d||^2$,
> This is a well-known optimization problem whose solution is directly obtained via the Courant–Fischer theorem, establishing that the optimal choice of $B$ consists precisely of the eigenvectors corresponding to the top c eigenvalues of $G$. We will provide a clearer, step-by-step derivation of this result in the revised appendix.
>
> **Q2.4: Modify the motivation to highlight the important task directions instead of magnitude issues.**
>
> **A2.4:** Thank you for this insightful suggestion. We will clarify our motivation to reflect that CAT Merging explicitly identifies and trims conflict-prone components based on their directional contributions to knowledge conflicts rather than relying solely on vector magnitudes.
>
> **Q2.5: Should not assume α=1.**
>
> **A2.5:** Thank you for raising this point. The assumption of α=1 was introduced solely to simplify the theoretical analysis, as the scaling factor α does not affect the resulting conclusion. For example, in Eqs. (9)&(10), if we explicitly include α, the optimal vector B corresponds to the eigenvector of the matrix:
> $\sum_{i\neq k} {(\alpha T_i)}^\top( {X_k}^\top X_k - \lambda {X_i}^\top X_i) {(\alpha T_i)}=\alpha^2\sum_{i\neq k} {T_i}^\top( {X_k}^\top X_k - \lambda {X_i}^\top X_i) {T_i}$.
> Since scaling a matrix by a nonzero constant α² does not alter its eigenvectors, our conclusions remain valid. Nevertheless, we agree with the reviewer that, in practical model merging, α is a critical hyperparameter that must be tuned carefully. We will provide necessary clarifications in the revised manuscript.
>
> **Q2.6: Isn't the upper bound vacuous in Theorem 5.1?**
>
> **A2.6:** We acknowledge that the bound presented in Theorem 5.1 is relatively loose. However, this bound is primarily intended to provide conceptual insight into the motivation underlying our layer-wise trimming strategy. Specifically, Theorem 5.1 establishes the inequality:
>
> $| L(W) - L( W + \Delta W) | < \beta \sum_{l=1}^L \Bigl(\prod_{m=l+1}^L \gamma_m\Bigr)
> \|\Delta f^l(W^l) - f^l(W^l + \Delta W^l) \|$.
>
> This inequality highlights that reducing layer-specific conflicts (the right-hand side) directly contributes to controlling the difference in model performance (the left-hand side). Thus, even though the bound itself is not tight, it provides theoretical justification for our subsequent conflict-aware trimming strategies at each layer.
>
> **Q2.7: Comparison of runtimes.**
>
> **A2.7:** Please see A 1.3 in our response to C9mN.
>
>
> **Q2.8: Figure 2 needs improvement.**
>
> **A2.8:** Thanks. We update Figure 2 in [sensitivity-experiment.png](https://postimg.cc/HjPTnbtt), which now more clearly demonstrates that our method remains stable in a rational range, with respect to both the number of exemplars and the value of α.
>
> **Q2.9: How are the exemplars selected in Figure 2?**
>
> **A2.9:** The exemplars are selected randomly, and we report the average performance of three random runs.
>
> **Q2.10: Do “shift” parameters mean biases?**
>
> **A2.10:** Yes. They include both the bias parameters of linear layers and the shift parameters in normalization layers. Since they share the same trimming form, we treat both of them as “shift” parameters for brevity.
>
> **Q2.11: Are attention layers treated as linear layers?**
>
> **A2.11:** We decompose it into several linear layers and treat them separately (e.g., three linear layers for Q/K/V calculations).
>
> **Q2.12: Writing issues (verbose formulations and citations).**
>
> **A2.12:** We will revise accordingly in the final version.

---

### Official Review · Reviewer_C9mN · 2025-03-14

**Overall Recommendation:** 3

**Summary:**

This paper proposes Conflict-Aware Task Merging (CAT Merging), a training-free method to combine multiple fine-tuned models while alleviating knowledge conflicts that degrade performance when merging. The core idea is to selectively trim conflict-prone components from each task’s weight update (“task vector”) instead of simply adding them. The approach applies parameter-specific strategies – projecting linear layer weight updates and masking normalization scale/shift parameters.

**Claims And Evidence:**

The claims made in the submission are generally supported.

**Essential References Not Discussed:**

I am not familiar with multi-task literature and therefore unaware of important references.

**Experimental Designs Or Analyses:**

The experiments look descent.

**Methods And Evaluation Criteria:**

The proposed methods and/or evaluation criteria (e.g., benchmark datasets) make sense

**Other Comments Or Suggestions:**

More comments.
- Figure 4(b) shows that the model performs best when c is relatively small (around 2 to 4). However, the trimmed task vector $TB_i^l (B_i^l)^\top$ lies within the column space of B, meaning that only a very low-dimensional subspace (2–4 dimensions) is being trimmed.  Does this suggest that knowledge conflicts are not as severe as initially claimed?
- Figure 4(a) shows that the model performs stable when lambda is relatively large. Since a small lambda places greater emphasis on inter-task knowledge conflict, we expect that the model’s performance should also be good if knowledge conflict is important. To fully understand the impact of the trade-off between knowledge preservation and conflict suppression, it would be more informative to include experiments where lambda falls into (0, 1).
- The value of Eq. (21) is exactly the top c eigenvalues of the matrix $G$. Could authors provide an energy plot that visualizes the eigenvalues of the matrix G? Such a plot would help assess whether the eigenvalues decay sharply or remain relatively large beyond c components.

**Other Strengths And Weaknesses:**

Strengths:
- The paper presents a novel solution to an important problem. This paper takes the concept of knowledge conflicts into consideration in the problem of task merging.
- The implementation of training-free conflict-aware merging is elegant. This work explicitly formulated and solved the dual-objective conflict minimization per layer.
- The experiment results are impressive and comprehensive.
Weakness:
- Over-prioritization vs magnitude. the paper mentions that differences in task vector magnitudes can lead to over-prioritization of certain task vectors, resulting in problematic merging. However, the proposed trimming method does not seem to fully resolve this issue. For example, for linear layers, the trimmed task vector is $T(I - B^l (B^l)^\top)$. The multiplication between the matrix $T$ and $B^l (B^l)^\top$ indicates that a task vector with a large magnitude before trimming will likely still have a large magnitude after trimming. As a result, the same over-prioritization problem mentioned in the Introduction could persist.
- High computation complexities. Although computing the trim matrix $B_k^l$ and $m_k^l$ does not require explicit training, it relies on eigenvector computation, which has a computational complexity of $O(n^3)$. This can be a significant bottleneck, especially when dealing with large-scale models where the hidden dimension can be extremely high (e.g., 4096).
- Activation and non-linear functions. All the formulas and implementations in this paper only consider the linear matrices. However, activation functions are an important part of neural networks, and this paper doesn’t discuss it at all. Could these activation functions can further simplify the trim matrices? For example, after ReLU, some parts of the outputs are deactivated, and we don’t need to consider the corresponding trim matrices.

**Questions For Authors:**

Aforementioned in previous sections.

**Relation To Broader Scientific Literature:**

This work situates itself at the intersection of multi-task learning and model merging, building upon and extending prior research in both domains. The authors provide a solid literature review (Section 2) that distinguishes traditional multi-task learning from the newer paradigm of model merging.

**Theoretical Claims:**

I have carefully reviewed all the theoretical formulations and proofs, and from my perspective, most of them appear to be correct.

However, I have concerns regarding Theorem 5.1. The theorem seems to assume that all linear layers are stacked sequentially, which is not always the case in practical architectures. For example, in Transformer attention layers, the mathematical formulation involves a multiplicative interaction:

$$z=σ(XW_Q (XW_K )^T )XW_V=σ(XW ^{KQ}X^T )XW_v,   W ^{KQ}=W_Q W_K^T$$

where three linear layers are multiplied together rather than applied in a purely sequential manner. This multiplicative structure makes the Lipschitz continuity assumption less reliable, as the final output’s dependence on input perturbations is quadratic or even cubic (at least $XW^{KQ}X^T$ is quadratic to the input). When the experiment results in this paper mainly focus on Transformer architectures, could the author explain this assumption further?

---

> ### Author Rebuttal · Authors · 2025-04-01
>
> **Q1.1: Is the Lipschitz continuity assumption becoming less reliable in Transformer architectures?**
>
> **A1.1**: We thank the reviewer for this insightful observation. Indeed, the multiplicative interactions in Transformer architectures complicate the Lipschitz continuity assumption. However, given that both the network parameters and the input data are practically bounded, we can still derive a sufficiently large Lipschitz constant, under which Theorem 5.1 remains valid—albeit with a looser upper bound. Importantly, we would like to emphasize that the primary role of Theorem 5.1 is to provide theoretical insight and motivation for our layer-by-layer trimming strategy, rather than deriving tight practical bounds. Please see response A2.6 for additional details.
>
> **Q1.2: Is the trimming method proposed to solve the magnitude issue?**
>
> **A1.2**: Not exactly. The proposed trimming method specifically addresses knowledge conflicts during model merging rather than magnitude issues alone. Magnitude-based techniques (e.g., Ties-Merging) attempt to resolve conflicts by masking low-magnitude components. However, as illustrated in Fig. 1, simply masking low-magnitude components does not fully eliminate conflicts, since high-magnitude components can also cause significant interference. In contrast, our CAT Merging explicitly identifies and trims components based on their actual contribution to knowledge conflicts, rather than simply their magnitude. This targeted strategy more effectively mitigates conflicts, leading to improved merging performance.
>
> **Q1.3: Does CAT Merging have a high computational complexity?**
>
> **A1.3:** The computational overhead of CAT Merging is reasonable and practically efficient. Specifically, CAT Merging involves two main steps:
>
> 1. **Feature Extraction:** This step is lightweight and efficient, requiring only a small number (2–3 per task) of unlabeled samples.
> 2. **Eigendecomposition:** While eigendecomposition has theoretically higher computational complexity, in practice, we efficiently mitigate this through GPU parallelization. Moreover, CAT Merging only requires the eigenvectors corresponding to the top-*c* (2-4 in our work) eigenvalues, enabling further acceleration through specialized methods (e.g., `torch.lobpcg`). Empirical results (provided in the table below, measured on a single RTX3090 GPU in seconds) demonstrate that CAT Merging performs much faster than training-based counterparts (e.g., TA w/ Surgery, AdaMerging).
>
> |  | ViT-B/32 | ViT-L/14 |
> | --- | --- | --- |
> | PCB Merging | 43 | 131 |
> | **CAT Merging (ours)** | 46 | 150 |
> | TATR | 176 | 283 |
> | TA w/ Surgery | 12621 | 36826 |
> | AdaMerging | 8276 | 16299 |
>
> **Q1.4: Could activation functions simplify the trimming matrices?**
>
> **A1.4:** We agree that a deeper analysis of activation functions would be beneficial. While CAT Merging does not explicitly model activation functions, it implicitly captures their effects through the layer-wise trimming strategy. For instance, activation functions such as ReLU can cause certain dimensions to consistently remain inactive. Referring to Section 5.2 of our paper, the trimming mask for scaling parameters is computed as:
> $$\sum_{i\neq k} \left( \sum_{x^l_k} ( x^l_k\circ T^l_i )^2 - \lambda \sum_{x^l_i}  ( x^l_i\circ T^l_i)^2 \right)$$
> If the $d$-th dimension remains consistently inactive (i.e., $x_k^l[d]  \equiv x_i^l[d] \equiv 0$), the corresponding element in the trimming mask naturally becomes zero. Thus, activation functions indirectly simplify the trimming process, even though they are not explicitly modeled in CAT Merging. We will clarify this point explicitly in our revised manuscript.
>
> **Q1.5: Why should only a very low-dimensional subspace (2-4 dimensions) be trimmed as suggested in Figure 4(b)?**
>
> **A1.5:** In Figure 4(b), trimming only a low-dimensional subspace (2–4 dimensions) is sufficient because knowledge conflicts are predominantly concentrated within a few principal dimensions. Specifically, as shown in [eign.png](https://postimg.cc/jCQwQn8Z), the first few eigenvectors represent the most significant directions, accounting for an average of 78.56% (ViT-B/32) and 87.28% (ViT-L/14) of the total eigenvalues. Thus, while conflicts are indeed severe, their severity primarily manifests along these critical dimensions.
>
> Moreover, trimming a higher-dimensional subspace risks unnecessarily degrading the original task performance, as it could remove important task-specific information. Therefore, selecting this low-dimensional subspace effectively balances conflict mitigation and preservation of model performance.
>
> **Q1.6: λ in Figure 4(a) should be tuned in the range of (0, 1).**
>
> **A1.6:** As suggested, we conducted additional experiments with λ values in the (0, 1) range. The results, provided in [sensitive-lambda.png](https://postimg.cc/xcsrvQkS), indicate that model performance consistently improves as λ increases from 0 toward 1, peaking around 5.

---

### Decision · Program_Chairs · 2025-05-01

**Decision:**

Accept (poster)

**Comment:**

This paper presents Conflict‑Aware Task Merging (CAT Merging), a training‑free method for combining multiple fine‑tuned models by selectively trimming conflict‑prone components of task vectors via layer‑wise projections for linear weights and masking for normalization parameters. Reviewers agree that the problem is important, the approach is novel and well‐motivated, and that the core claims are broadly supported by both theoretical derivations and empirical results on vision and vision‑language benchmarks, with gains over prior model‑merging techniques at effectively zero incremental memory cost. However, several concerns must be addressed: the theoretical analysis (in particular Theorem 5.1) relies on assumptions (e.g., sequential linear layers, Lipschitz continuity) that may not hold for architectures with multiplicative interactions such as Transformers, and some proofs omit constructive arguments for choosing the trimming dimension; the computational cost of eigenvector computations at high hidden dimensions, the omission of activation functions from the analysis, and the lack of runtime comparisons need clarification; the presentation would benefit from clearer notation, inclusion of all baselines (especially in the final table), lambda‑tradeoff experiments, energy plots of eigenvalue decay, and broader evaluation on LLM benchmarks; and minor writing fixes and expanded discussion of suboptimal cases are needed. Given the paper’s solid foundation but these outstanding issues, I recommend a weak accept.